# Common schizophrenia risk variants are enriched in open chromatin regions of human glutamatergic neurons

Mads E. Hauberg [1,2,3,4,5,13], Jordi Creus-Muncunill[6,13], Jaroslav Bendl [1,2,7,13], Alexey Kozlenkov[1], Biao Zeng[7], Chuhyon Corwin [6,7,8], Sarah Chowdhury[6,7,8], Harald Kranz[9], Yasmin L. Hurd [1,2,10], Michael Wegner [11], Anders D. Børglum [3,4,5], Stella Dracheva[1,2,12], Michelle E. Ehrlich[6,7,8], John F. Fullard[1,2,7] & Panos Roussos [1,2,7,12✉]

The chromatin landscape of human brain cells encompasses key information to understanding brain function. Here we use ATAC-seq to profile the chromatin structure in four distinct populations of cells (glutamatergic neurons, GABAergic neurons, oligodendrocytes, and microglia/astrocytes) from three different brain regions (anterior cingulate cortex, dorsolateral prefrontal cortex, and primary visual cortex) in human postmortem brain samples. We find that chromatin accessibility varies greatly by cell type and, more moderately, by brain region, with glutamatergic neurons showing the largest regional variability. Transcription factor footprinting implicates cell-specific transcriptional regulators and infers cell-specific regulation of protein-coding genes, long intergenic noncoding RNAs and microRNAs. In vivo transgenic mouse experiments validate the cell type specificity of several of these human-derived regulatory sequences. We find that open chromatin regions in glutamatergic neurons are enriched for neuropsychiatric risk variants, particularly those associated with schizophrenia. Integration of cell-specific chromatin data with a bulk tissue study of schizophrenia brains increases statistical power and confirms that glutamatergic neurons are most affected. These findings illustrate the utility of studying the cell-type-specific epigenome in complex tissues like the human brain, and the potential of such approaches to better understand the genetic basis of human brain function.

[1] Department of Psychiatry, Icahn School of Medicine at Mount Sinai, New York, NY 10029, USA. [2] Friedman Brain Institute, Icahn School of Medicine at Mount Sinai, New York, NY 10029, USA. [3] iPSYCH, The Lundbeck Foundation Initiative for Integrative Psychiatric Research, Aarhus, Denmark. [4] Department of Biomedicine, Aarhus University, Aarhus, Denmark. [5] Centre for Integrative Sequencing (iSEQ), Aarhus University, Aarhus, Denmark. [6] Department of Neurology, Icahn School of Medicine at Mount Sinai, New York, NY 10029, USA. [7] Department of Genetics and Genomic Science and Institute for Multiscale Biology, Icahn School of Medicine at Mount Sinai, New York, NY 10029, USA. [8] Department of Pediatrics, Icahn School of Medicine at Mount Sinai, New York, NY 10029, USA. [9] Gene Bridges, Im Neuenheimer Feld 584, 69120 Heidelberg, Germany. [10] Department of Neuroscience, Icahn School of Medicine at Mount Sinai, New York, NY 10029, USA. [11] Institut für Biochemie, Emil-Fischer-Zentrum, Friedrich-Alexander Universität Erlangen-Nürnberg, Erlangen, Germany. [12] Mental Illness Research, Education, and Clinical Center (VISN 2 South), James J. Peters VA Medical Center, Bronx, NY, USA. [13]These authors contributed equally: Mads E. Hauberg, Jordi Creus-Muncunill, Jaroslav Bendl. ✉email: panagiotis.roussos@mssm.edu

Cell-type-specific variations in the epigenetic regulation of gene expression are critical to the development and maintenance of a healthy human brain. As most disease- and trait-associated variants affect the epigenetic regulation of gene expression rather than protein-coding sequence[1], studying these processes at the cell-type-specific level is an important means to further understand both fundamental brain biology and the genetic basis of neuropsychiatric disease.

Despite its functional importance, the epigenome of the human brain is still poorly understood. Even when available, the isolation of intact cells from fresh brain specimens is technically challenging and, although promising, the use of iPSC-derived brain cells or organoids are not ideal proxies. Frozen archival tissue is more readily available; however, the majority of cell surface markers are lost upon thawing and, with them, an important means to isolate particular cells of interest. Due to a shortage of antibodies specific to nuclei of a given cell type, most previous studies have been limited to examining bulk tissue, in vitro cultured cells, have included only two broadly defined brain cell types (neurons and non-neurons), or were performed on tissues derived from a single brain region[2–4].

Previously, we generated an atlas of chromatin accessibility in the human brain[5] using an anti-NeuN (RBFOX3) antibody to distinguish neuronal from non-neuronal nuclei[6]. In the current study, we expand the panel of antibodies used by including SOX6, to distinguish GABAergic (GABA) (NeuN+/Sox6+) from glutamatergic (GLU) (NeuN+/Sox6−) neurons[7], and SOX10, to distinguish oligodendrocytes (OLIG) (NeuN−/Sox10+) from microglia/astrocytes (MGAS)[8] (NeuN−/Sox10−). Nuclei isolated in this manner from three different brain regions—anterior cingulate cortex (ACC), dorsolateral prefrontal cortex (DLPFC), and primary visual cortex (PVC)—are subjected to ATAC-seq and the resulting data are used to explore differences in chromatin structure at the level of cell type and brain region. This approach yields insights to differences in biological function and gene regulation, and provides a map of open chromatin to genetic variants associated with neuropsychiatric traits. We provide our raw data and gene browser tracks to the scientific community.

## Results

### Extensive chromatin accessibility in glutamatergic neurons.
Fluorescence-activated nuclear sorting[4] (FANS) followed by ATAC-seq was used to determine chromatin accessibility in four cell types (GABA, GLU, OLIG, and MGAS) across three brain regions (ACC, DLPFC, and PVC) of four individuals in early adulthood (ages 20–28), who had not been diagnosed with neuropsychiatric illness at the time of death, and all with a postmortem interval less than 24 h (Fig. 1a and Supplementary Fig. 1). These data were processed bioinformatically including various checks for sample mix-ups (Supplementary Fig. 2). During quality control, one ACC-derived GABA sample was found to be of poor quality as it showed a low final read count, a low fraction of reads in peaks and was an outlier in a clustering analysis. After removing this sample, the remaining 47 samples yielded a total of 3.10 billion reads (average 65.9 million) excluding duplicate reads (mean 25.2%) and reads mapping to the mitochondrial genome (mean 1.32%) (Supplementary Data 1). Genome-wide correlations of read counts between samples indicated a high reproducibility (Supplementary Fig. 3a), and our samples showed a median TSS read count enrichment of 4.2, comparable to similar studies (Supplementary Fig. 3b).

Open chromatin regions (OCRs) were called after merging samples from the same brain region and cell type (Methods). A total of 177,178 non-overlapping OCRs were detected, jointly covering 2.89% of the genome, with the highest coverage seen in the GLU (Fig. 1b). On average, 62.4% of the OCRs were identified in two or more sample groups, with promoter OCRs most frequently called in multiple sample groups compared to non-promoter OCRs (Fig. 1c). On average, 30.2% of the OCRs were promoter OCRs (Fig. 1d, e), but this varied by cell type with the lowest fraction in GLU, demonstrating more distal regulation in those samples. Thus, GLU showed more extensive tracts of open chromatin than other cell types, suggesting more complex regulation of gene expression. To assess the function of these OCRs we tested for enrichment in genetic variants affecting gene expression in GTEx eQTLs[9,10]. All cell types showed enrichment in credible genetic variants (odds ratios: 1.6–2.9; empirical $p <$ 0.05; Supplementary Table 1).

### Cell type and regional differences in chromatin structure.
To quantitatively assess differences in chromatin structure between cell types and brain regions, we generated a 177,152 OCRs by 47 samples count matrix for ATAC-seq reads that overlap the consensus peak set. We examined how biological and technical covariates affect chromatin accessibility (Methods; Fig. 1f). Cell type and brain region jointly account for 57% of variability. Fraction of reads in peaks was the only technical covariate selected in a stepwise Bayesian information criterion approach, accounted for 7.2%. In a model not accounting for brain region, cell type alone accounted for 54% of the variance (Supplementary Fig. 4), indicating that, compared to brain region, cell type has a larger effect size on chromatin accessibility. Sampling error from finite sampling depth and untested technical confounds likely contribute to the residuals in the model. This includes covariates relating to the individual person such as postmortem interval, but as such covariates would show collinearity with "Person" in the model, which explains only a modest fraction of the variance. Such person-related covariates are unlikely to drastically affect chromatin variability.

The count matrix was then adjusted for technical confounders and used for t-SNE clustering (Fig. 1g), yielding a clear separation between the cell types with the most marked difference between neurons and non-neurons and, to a lesser extent, between cellular subtypes (GLU from GABA and OLIG from MGAS). To further quantify the differences in chromatin accessibility between different groups of samples, we grouped the samples by cell type and brain region (e.g. one group consisted of the four DLPFC-derived OLIG samples). We then quantified the statistical difference between all group pairwise comparisons by calculating *pi1*, which is an estimate of the fraction of OCRs showing a difference in chromatin accessibility between the two sample groups. The neuronal vs non-neuronal samples showed the highest *pi1* (median = 72.3%, standard deviation = 5.80%) (Fig. 1h) followed, in decreasing order, by comparisons between OLIG and MGAS, between GLU and GABA, and regional differences between samples of a given cell type. Based on the *pi1* estimate, GLU showed the greatest regional variation in chromatin accessibility.

To identify cell-specific OCRs, we conducted analyses of differential chromatin accessibility in the four individual cell types (GLU, GABA, OLIG, and MGAS) as well as more broadly defined cell types (neuronal and non-neuronal). For the individual cell types, OCRs were considered specific to a cell if they were significantly more accessible in all pairwise comparisons against the remaining three cell types, thus yielding a non-overlapping set of cell-specific OCRs. A similar approach was used to define non-overlapping OCRs specific to neuronal and non-neuronal samples (Methods). This yielded OCRs specific to GLU (38,531), GABA (17,751), OLIG (11,030), MGAS (18,834),

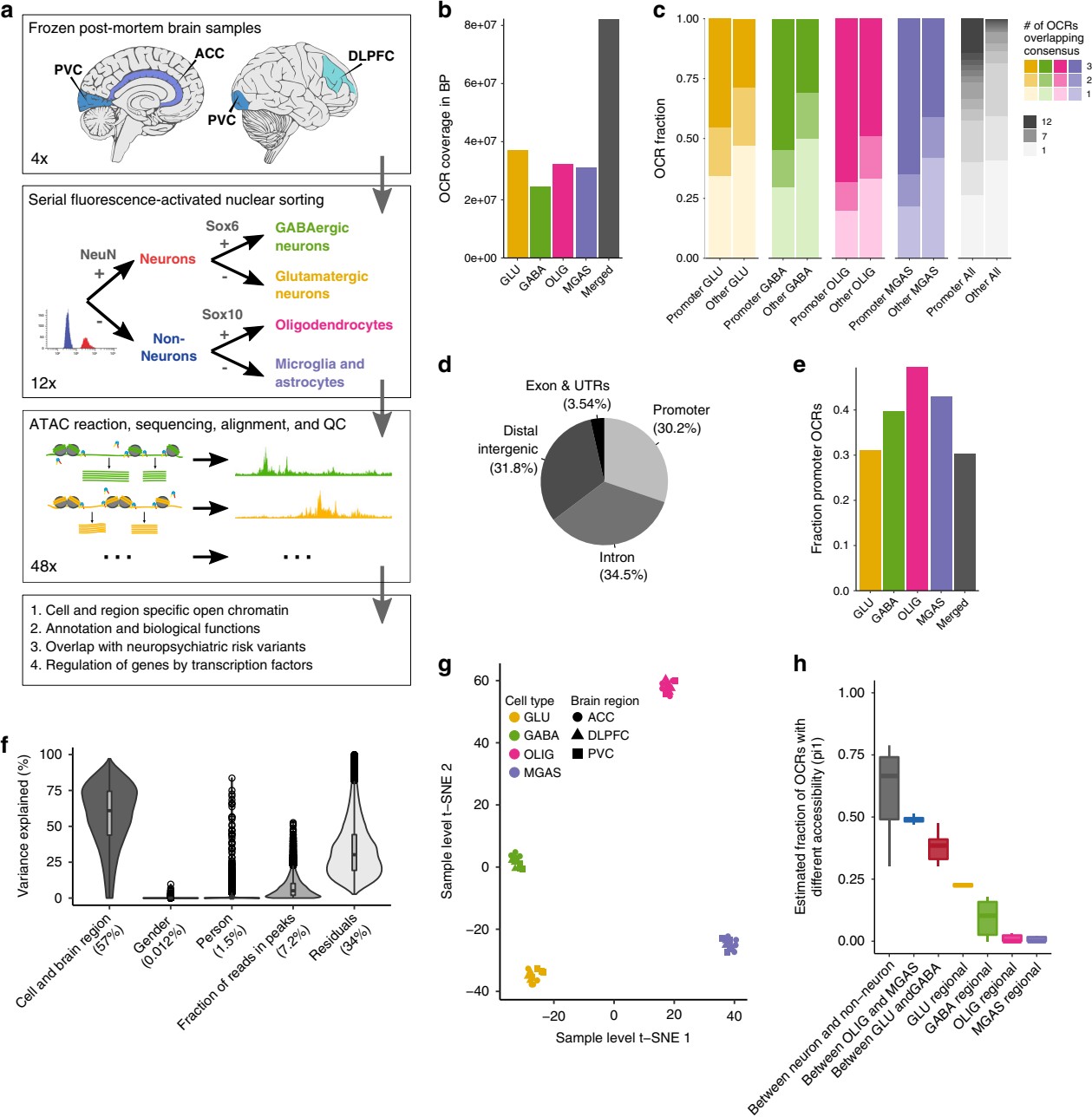

**Fig. 1 Outline of study and chromatin accessibility across four cell types. a** Study design: Dissections from three brain regions of four early adulthood control subjects were obtained from frozen human postmortem tissue (ACC: anterior cingulate cortex; DLPFC: dorsolateral prefrontal cortex; and PVC: primary visual cortex). Nuclei were subjected to fluorescence-activated nuclear sorting to yield four-cell subpopulations, followed by ATAC-seq profiling and subsequent downstream analyses to identify cell type-specific open chromatin regions and differences in biology. **b** Genomic coverage in base pairs of identified OCRs by cell type. **c** Number of OCRs called per sample groups. For the individual cell types three overlaps mean that the OCR was detected in all three brain regions and for "All" 12 overlaps means the OCR was detected in all brain regions and cell types. **d** Genomic annotation of consensus OCRs. OCRs within 3 kb of a TSS were considered promoter OCRs. **e** Fraction of OCRs considered as promoter OCRs by cell type. **f** Violin plot that illustrated the proportion of variation in chromatin accessibility explained by biological and technical covariates. The fraction of reads in peaks can be considered a signal to noise parameter. Numbers in parentheses indicate median. **g** t-SNE clustering of chromatin accessibility using adjusted read counts in 47 independent samples. **h** Quantification of statistical differences between various cell types by brain region comparisons using the pi1 metric. The center shows the median, the box shows the interquartile range, whiskers indicate the highest/lowest values within 1.5x the interquartile range, and potential outliers from this are shown as dots. From left to right, the number of independent contrasts represented by each boxplot are: 54, 9, 9, 3, 3, 3, and 3. The pi1 estimates the proportion of non-null tests. The boxplot shows the pi1 estimate between the relevant sample groups. For instance, "Between OLIG and MGAS" are all pairwise comparisons of the three OLIG sample groups (ACC/DLPFC/PVC) and the three MGAS sample groups (ACC/DLPFC/PVC). OCR open chromatin region, GLU glutamatergic neurons, GABA GABAergic neurons, OLIG oligodendrocytes, MGAS microglia and astrocytes, ACC anterior cingulate cortex, DLPFC dorsolateral prefrontal cortex, and PVC primary visual cortex.

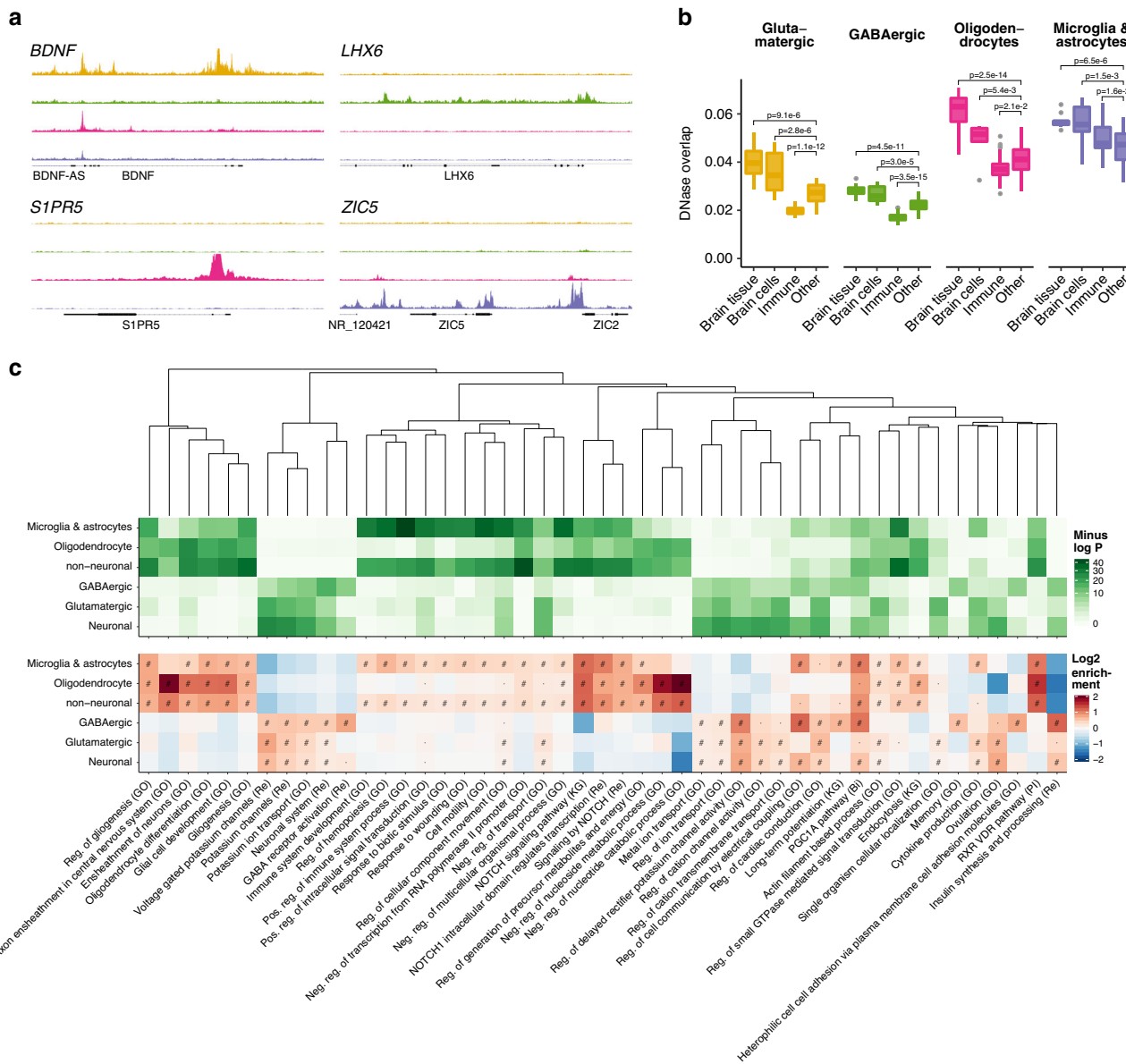

**Fig. 2 Cell-specific OCRs, overlap with DNAse-seq, and biological functions. a** Examples of genes with cell-specific open chromatin. Cell types from top to bottom are; glutamatergic neurons, GABAergic neurons, oligodendrocytes, and microglia/astrocytes. **b** Overlap between cell-specific open chromatin (ATAC-seq) and 127 samples from REMC (DNAse-seq). The overlap was calculated by the Jaccard index of the base pair overlap. Samples from REMC were aggregated into four groups: brain tissue, brain-derived cells, immune cells/tissues, and other non-brain cells/tissues. The center shows the median, the box shows the interquartile range, whiskers indicate the highest/lowest values within 1.5x the interquartile range, and outliers from this are shown as dots. The number of independent sample overlaps represented by the boxplot groups are as follows: Brain tissue: 10, Brain cells: 6, Immune: 30, and Other: 81. To assess the significance of the differences in overlap for our cell-specific OCRs with the four REMC categories, a multiple regression analysis with the "Other" category as the intercept was done. *P*-values indicate significance of enrichment/depletion against the other category uncorrected for multiple testing. **c** Overlap between cell-specific open chromatin (ATAC-seq) and gene sets representing biological processes and pathways. Only those that were within the top ten most significant gene sets in one or more ATAC-seq categories are shown. Pathways were clustered by the Jaccard index using the WardD method. "#": one-sided binomial FDR < 0.001, "·": one-sided binomial FDR < 0.05, "Bi": BIOCARTA, "GO": gene ontology, "KG": KEGG, "Re": REACTOME, "Reg.": regulation, "Pos." positive, "Neg." negative.

neurons (105,550), and non-neurons (34,282) at a false discovery rate (FDR) of 5% (Supplementary Fig. 5; Supplementary Data 2). The cell specificities identified here were highly concordant with other ATAC-seq and RNA-seq studies (Supplementary Fig. 6). We show examples of regions harboring cell-specific open chromatin in Fig. 2a. Analyzing regional OCR differences in each cell type yielded statistically significant differences only for GLU (Methods). Here, 258 OCRs showed a higher accessibility in ACC, 2807 in DLPFC, and 770 in PVC (Supplementary Data 3).

For downstream analyses, we analyzed the cell and region-specific OCRs as well as all OCRs discovered in a given cell type (e.g. including non-specific OCRs), due to the complementary nature of these approaches.

**Biological interpretation of open chromatin in brain cells.** To investigate how our data compared to existing epigenomic data, we computed the overlap of our OCRs with open chromatin from

**Table 1 Cell-specific transcripts identified from open chromatin.**

| Cell type | Protein coding | lncRNA | microRNA |
|---|---|---|---|
| Neuronal | 4075 | 3417 | 87 |
| Glutamatergic | 807 | 1123 | 8 |
| GABAergic | 834 | 698 | 12 |
| non-Neuronal | 6289 | 4485 | 89 |
| Oligodendrocyte | 2318 | 1386 | 25 |
| Microglia/astrocytes | 1887 | 1957 | 33 |

To infer the transcriptome from chromatin accessibility, genes were linked to cell-specific open chromatin regions by direct overlap of the TSS. "Neuronal" is not simply the sum of "Glutamatergic" and "GABAergic", as the latter two exclude chromatin that are not specific to either neuronal subtype (Methods). Likewise, "non-Neuronal" is not the sum of the two constituent cell subtypes. For microRNA host genes, only genes encoding one or more conserved microRNA were considered.

DNase-seq as well as chromatin states from the Roadmap epigenomics mapping consortium[2,11] (REMC) (Fig. 2b and Supplementary Fig. 7). In terms of the Jaccard index, open chromatin and active chromatin states identified in REMC brain-related samples showed a higher overlap with our cell-specific OCRs than non-brain related samples. Comparing to the genomic background, our cell-specific OCRs showed 5-27 fold enrichments in the brain related REMC DNase samples (Supplementary Fig. 7b). Overall, the OLIG and MGAS specific OCRs showed the highest overlap with REMC open chromatin, indicating that studying bulk tissue might be less effective at capturing GLU and GABA specific open chromatin than their non-neuronal counterparts. Interestingly, the MGAS specific OCRs showed a comparatively high overlap with open chromatin of immune-related samples, likely due to the myeloid origin of microglia and the extent of their functional similarity[12].

To examine the overlap of OCRs with previously reported cell type and brain region-specific markers and genes involved in biological processes, we implemented the gene set enrichment analysis methodology from GREAT[13] (Methods). Here, the cell-specific OCRs were found to overlap relevant cell-type-specific genes[14,15] (Supplementary Fig. 8) with, for example, GABA OCRs primarily overlapping interneuron specific genes. Similarly, our cell-specific OCRs showed enrichment in genes previously identified from bulk tissue to show brain region specificity[16] (Supplementary Fig. 9). It is possible that the ability to identify brain region-specific genes in bulk tissue is affected by differences in cell-type composition (e.g. the ratio of neuronal to non-neuronal cells). Finally, cell-type-specific OCRs overlapped genes of relevant biological functions (Fig. 2c and Supplementary Fig. 10) with, for example, "Voltage Gated Potassium Channels" in GLU; "GABA Receptor Activation" in GABA; "Axon Ensheathment in Central Nervous System" in OLIG; and "Immune System Development" in MGAS. For the comparatively few regionally specific GLU OCRs, enrichments were found for mostly plausible pathways, though none survived multiple testing (Supplementary Fig. 11).

**Inferring protein, lncRNA, and microRNA activity.** Isolating human brain cells for transcriptomic studies is challenging and, although nuclear isolation from frozen specimens is possible (as in this study), in those cases, the cytoplasmic transcriptome is lost. Despite these challenges, inferring the complete cell-specific transcriptome from the epigenome is of great interest as it has the potential to yield biological insights, including for rare and unstable transcripts. In an effort to achieve such an inference, we overlapped the OCRs with TSSs of microRNA[17], lncRNA[18], and protein-coding genes, and evaluated cell specificity from the relative accessibility in the four cell types. This approach revealed thousands of cell-specific genes (Table 1 and Supplementary Data 4). This cell-specific map of microRNAs includes mir-219a-2 and mir-338 as the most OLIG specific genes, both of which have previously been shown to play a central role in oligodendrocyte development and function[19]. No microRNA showed extreme specificity to GLU or GABA but the TSSs ascribed to mir-129-2 and mir-133a-1 showed accessibility predominantly in the former, whereas those for mir-23c and mir-124-1 dominated in the latter. We also note that top lncRNAs in our mapping plausibly overlap with cell/tissue specificity in the datasets from which they were derived18. Finally, the protein-coding gene map showed cell-specific enrichments in biological functions similar to those identified by the GREAT analysis (data not shown). This alternative approach to gene set enrichment analyses, however, seemed less powerful.

**Cell-type-specific transcription factor activities.** To interrogate gene regulation in the different samples we conducted transcription factor (TF) footprinting analyses[20] to predict binding within the identified OCRs. We used 431 TF motifs that were aggregated from a meta-database[21] and jointly represented 807 TFs (Methods). As an estimate of the impact of each transcription factor on regulating a gene, we weighed each TF binding sites by the distance to the TSS and the probability of the binding site being bound. We next performed pairwise comparisons of transcript regulation using the mean rank (Methods) for protein-coding genes (Fig. 3a), lncRNAs (Supplementary Fig. 12), and microRNAs (Supplementary Fig. 13). For protein-coding genes, this highlighted, amongst others, *BDNF* in GLU; *DLX6* in GABA; *SOX8* in OLIG; and *ZIC5* in MGAS. Interestingly, many of the top-ranking GABA-associated genes were members of the DLX family of homeodomain transcription factors which are known to play important roles in the development and function of interneurons[22].

To identify the transcription factors that potentially mediate cell-specific gene regulation, we calculated the overlap of footprinted transcription factor binding sites with cell-specific OCRs and compared this to the overlap with OCRs specific to the other cell types (Fig. 3b). It should be noted that there is extensive sharing of binding motifs between transcription factors within transcription factor families[21], so it is often not possible to determine which TF(s) of a TF family binds to a given binding site. Still, there is previous evidence to support some of the identified TF/cell associations. In particular, we note that: the bZIP TFs Jun/Fos jointly form the AP1-complex which is important for neuronal function[23]; the RFX and IRX families of TFs are further involved in neuronal function/development[24,25]; the bHLH TFs are important for neurogenesis[26], and we note that *NEUROD6* shows chromatin accessibility and regulation highly specific to GLU (Fig. 3b); the homeodomain TFs were associated with neurons and LHX6/8, in particular with GABA. Interestingly, *LHX6* shows highly specific chromatin accessibility in GABA and has been shown to regulate interneuron migration[27]; finally, the Ets family of TFs, though involved in a broad range of activities, have been shown to be important for microglia function[28].

**Evaluation of putative cell-type-specific enhancers.** Next, we wanted to validate the enhancer activity for a subset of cell-specific OCRs. Putative enhancer sequences for each cell type were selected if: 1) they showed cell-type specificity, 2) they were distal to TSS for a transcript that had chromatin accessibility in the TSS for the given cell type, 3) they were near to only one gene, 4) the nearby transcript had literature support for expression in the corresponding cell type. This resulted in putative

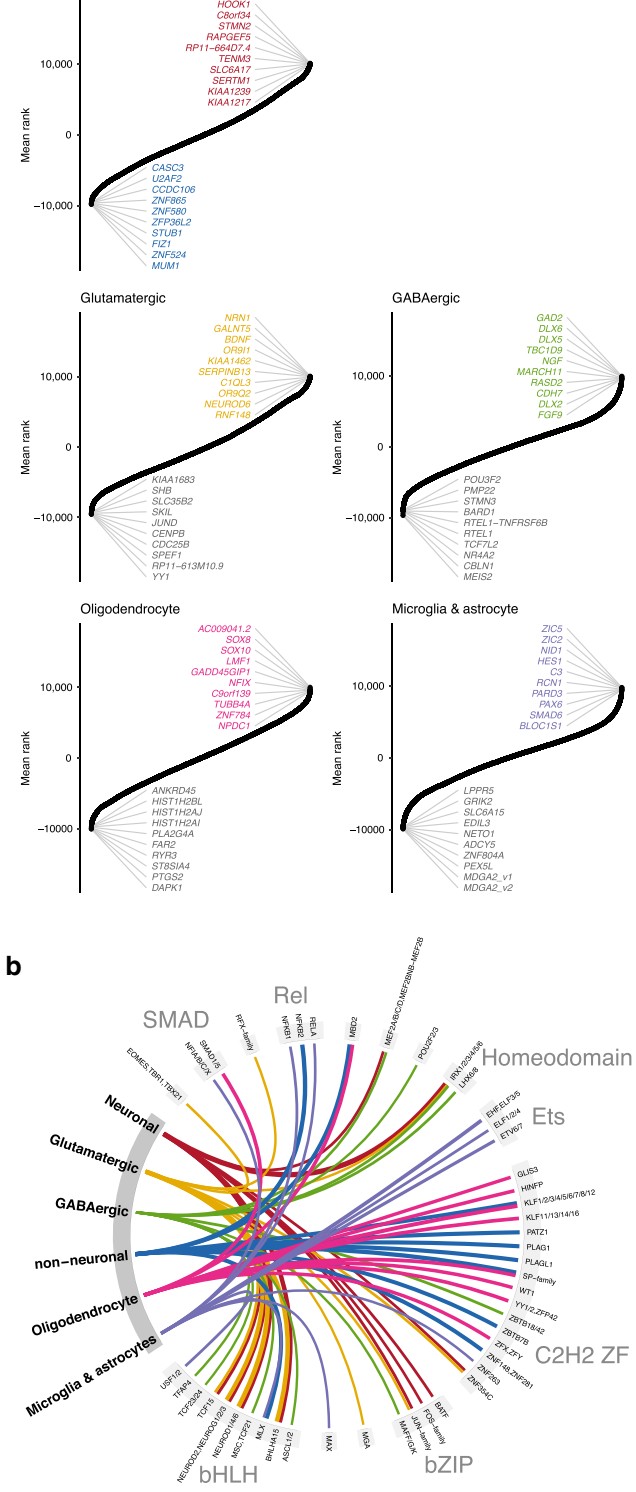

**Fig. 3 Gene regulation inferred from genomic footprinting. a** Identification of protein-coding genes showing cell-specific regulation. Aggregated ranking of pairwise comparisons of protein-coding gene regulation between neuronal/non-neuronal cells and each of the four different cell types. In neuronal/non-neuronal cell comparison, positive and negative values indicated a higher burden of gene regulation in neuronal and non-neuronal samples, respectively. For the GABA, GLU, OLIG, and MGAS analyses, each of the four different cell types was compared to the remaining three cell types. Positive values indicated a higher burden of gene regulation in the given cell type and negative value a lower burden of gene regulation than in the other cell types. **b** Top 10 most cell-specific TF motifs based on fold enrichment in cell-specific OCRs. Fold enrichments for a given cell type were determined from the number of footprinted binding sites overlapping cell-specific OCRs compared to the number of footprinted binding sites overlapping OCRs specific to the other cells. The statistical significance of the enrichments was assessed using a one-sided binomial test. All associations illustrated here were significant after Bonferroni corrections for multiple testing. The line width indicates the log2 fold enrichment. Motifs are grouped based on the TF family to which they belong.

and in the hippocampus (Fig. 4a and Supplementary Fig. 15), which are known sites of endogenous BDNF expression[29]. For the DLX6 OCR, 40% of the mCherry(+) cells were positive for GABA, and in all but one founder, mCherry was seen exclusively in cells expressing the neuronal marker, NeuN. These included variable percentages of neurons double-positive for mCherry, and either somatostatin (Sst), neuropeptide Y (Npy), parvalbumin (Parv), and vasoactive intestinal peptide (Vip) (Fig. 4b and Supplementary Fig. 16). Comparatively few founder mice were positive for the mCherry reporter (6/11), and a considerable variability in the expression pattern between animals was noted. For the CNDP1 OCR, All mCherry(+) cells were positive for the oligodendrocyte marker Olig2, and approximately 40% of the oligodendrocytes were mCherry(+) in the cortex and corpus callosum (Fig. 4c and Supplementary Fig. 17). Finally, for the TYROBP OCR, many mCherry(+) cells unexpectedly expressed the neuronal marker NeuN, and were negative for microglial and astrocytal markers (Iba1 and GFAP, respectively) (Supplementary Fig. 18). In summary, the putative GLU and OLIG enhancer candidates showed activity with the expected regional and cell-type specificity. For the putative GABA enhancer, activity was seen in GABAergic neurons, but not exclusively, and the putative microglial enhancer unexpectedly displayed some activity in neurons.

**Common genetic variants show cell-type-specific enrichment.** Given that most disease- and trait-associated genetic variants affect the gene regulation of gene expression rather than protein structure[1], we used an LD-score partitioned heritability approach[30] to explore the overlap of OCRs with genetic variants associated with 30 neuropsychiatric and unrelated traits, while correcting for the genomic background (Fig. 5a and Supplementary Fig. 19). Among our results, we found that variants associated with SCZ, education years and intelligence were enriched in OCRs specific to GLU. As a negative control, genetic variants associated with inflammatory bowel disease, height, and coronary artery disease did not show any enrichment. Some neuropsychiatric traits did not show enrichment in our open chromatin regions, which might result from a lack of power in the GWAS, lack of power in the LDSc approach, or the limited genomic extent of our epigenomic annotations.

To further explore the overlap with SCZ risk, we looked at the regression coefficient normalized by the per-SNP heritability of the trait (Methods) for different sets of OCRs. From this study, and our previous study of multiple brain regions[5], we considered

enhancer sequences for BDNF (GLU), DLX6 (GABA), CNDP1 (OLIG), and TYROBP (MGAS) transcripts.

These sequences were then evaluated using mouse transgenesis (Methods, Supplementary Fig. 14, Table 2). Here 15–22% of live pups born carried the transgene, at least five transgene positive animals were evaluated for each putative enhancer and, on average, 68% showed reporter gene activity (mCherry).

For the BDNF OCR, all mCherry(+) cells expressed the neuronal marker NeuN, and the majority of the cells were Bcl11b(+). The majority of immunopositive cells were in layer V of the cortex

**Table 2 Validation of putative enhancer regions by mouse transgenesis.**

| Putative enhancer region | Founders/% positive | Reporter gene | Expression pattern | Cell/region specificity |
|---|---|---|---|---|
| **BDNF** *glutamatergic* | 6/14.6% | 5/6 | 5/5 identical | Neurons in cortical layer V and hippocampus |
| **DLX6** *GABAergic* | 11/21.2% | 6/11 | Variable | Neurons, 40%+ GABAergic, variable subtypes |
| **CNDP1** *oligodendrocytes* | 6/22.2% | 4/6 | 4/4 identical | Oligodendrocytes |
| **TYROBP** *microglia/astrocytes* | 10/20.8% | 4/5 | 3/4 identical | Neurons |

Four open chromatin regions showing cell specificity in the human brain were evaluated for enhancer activity in mice, here named by their putative target genes. The cell specificity in humans of the open chromatin region is shown in the leftmost column. Genomic locations of the OCRs are provided in Methods. mCherry was used as a reporter. All positive founders were evaluated except for TYROBP where 5/10 were evaluated.

both cell type and cell/region-specific OCRs. We also included two studies that utilized bulk, unsorted, tissue samples from fetal and adult human tissue[31,32]. For SCZ (Fig. 5b), the signal seems to almost exclusively be neuronal and, among neurons, GLU-specific OCRs showed the highest enrichment. Bulk DLFPC and fetal brain also showed a significant, albeit less marked, enrichment than the GLU-specific OCRs. For Alzheimer's disease (AD) (Fig. 5c), we saw no enrichment in any of the neuronal or bulk epigenomes, whereas there was a tendency for enrichment in those of non-neurons. Specifically, heritability for AD seems to be very highly enriched in microglia- and astrocyte-specific open chromatin, albeit at nominal significance, consistent with a purported causative role of microglia in the trait[33,34]. Jointly, this highlights the importance of studying tissue- and cell-type-specific gene regulation to properly interrogate epigenomic disease signatures.

**The schizophrenia epigenome implicates glutamatergic neurons.** Given that schizophrenia risk genetic loci were significantly enriched within GLU-specific OCRs, we next explored whether chromatin accessibility was altered in cases with schizophrenia compared to controls. For this, we employed a recently published chromatin accessibility analysis from the CommonMind Consortium[31,35] of homogenate DLPFC tissue from 121 cases with schizophrenia and 126 controls. We note that the original study did not consider correction for heterogeneity in cell-type composition and it did not reveal disease signatures in chromatin accessibility[31]. We examined the utility of our cell-specific dataset as a reference panel to perform deconvolution analysis and to improve the power for differential analysis in the homogenate study by first removing heterogeneity due to differences in cell type composition (Methods, Supplementary Data 5). We identified individual heterogeneity in cell-type composition, which was not significant among cases and controls, suggesting that differences in cell composition among samples may be due to dissection biases (e.g. different ratio of gray to white matter) rather than disease effects (Supplementary Fig. 20). The selected cell type estimates that we used as covariates in the differential analysis among schizophrenia cases and controls explained overall 10.9% of variation in chromatin accessibility (Fig. 6a; Methods) and increased statistical power to identify disease signatures (Fig. 6b). The statistical difference between schizophrenia and controls estimated by *pi1* was increased by 52% when cell composition was considered (*pi1* = 7.3%) compared to the model where it was not (*pi1* = 4.8%). We then examined whether OCRs for a given cell type showed a higher statistical difference between SCZ and controls. We found that GLU-specific OCRs showed the highest statistical difference (*pi1* = 7.6%) (Fig. 6b), with a 24% increase in the *pi1* estimate compared to GABA (*pi1* = 6.3%), OLIG (*pi1* = 5.7%) and MGAS (*pi1* = 6.4%). Overall, we increased the power in a homogenate study by utilizing our reference OCR panel and identified GLU as the most impaired in schizophrenia with respect to chromatin accessibility.

## Discussion

Understanding spatial differences in epigenome regulation of the human brain tissue is a major challenge due to heterogeneity in cell-type composition. Here we examined cell type and brain region variability of chromatin accessibility by generating ATAC-seq libraries in four broadly defined cell types across three cortical brain regions. Based on clustering and differential analyses, we found chromatin accessibility to vary markedly by cell type and moderately by brain region. For instance, an oligodendrocyte from the DLPFC is very similar to one from the PVC but vastly different from a GABAergic neuron. Among all cell types studied, glutamatergic neurons showed the largest regional variation in chromatin structure and had the highest fraction of chromatin accessible regions that were distal to TSS, suggesting regulatory mechanisms with higher complexity. A larger sample size would probably lead to identification of more regional variation, but based on the variance partitioning and *pi1* estimates will be much subtler than cell-type differences.

We linked OCRs to transcripts by direct overlap with their TSS as well as by inferring transcription factor regulation. This allowed us to predict cell-specific transcript expression and regulation for protein-coding genes, as well as lncRNA and microRNA, at cellular resolution. We acknowledge that this is an indirect way to catalog cell-specific transcripts; however, other approaches, such as nuclear transcriptome profiling, suffer from a variety of limitations such as detection bias for transcripts with lower abundance, including lncRNAs.

Our ATAC-seq experiments identified cell and spatially specific OCRs, some of which represent enhancer sequences. Functional validation of such genome-wide enhancer candidate maps are needed, but few such studies are available. In an investigation of retinal rods and cones[36], validation via electroporation was as low as ~25%. In this study, we improved the efficacy of functional validation through transgenesis by including enhancer-blocking insulators to prevent interactions with chromatin surrounding the transgene insertion site. This led to successful validation of enhancer activity in 75% of OCRs tested.

Possible explanations for the lack of validation for the MGAS (TYROBP) OCR include: (1) interspecies differences; (2) specific promoter-enhancer interactions are lost when only transfecting the putative enhancer; (3) conformational/insulator differences between the in situ enhancer and the transgenic mice; (4) uncertainty of the original OCR function; and (5) insertional effects overcame the CTCG insulators in the vector. Overall, our results argue that coupling ATAC-seq with functional validation through transgenesis provides a powerful means to identify cell-specific enhancer activity. These enhancer sequences can be used as drivers for molecules such as EGFP and Cre recombinase to trace specific neuronal connections, suppress the expression of specific genes, and modulate activity via optogenetics and chemogenetics.

Common risk variation for complex, neuropsychiatric traits are located within noncoding regions of the genome. Overlap of

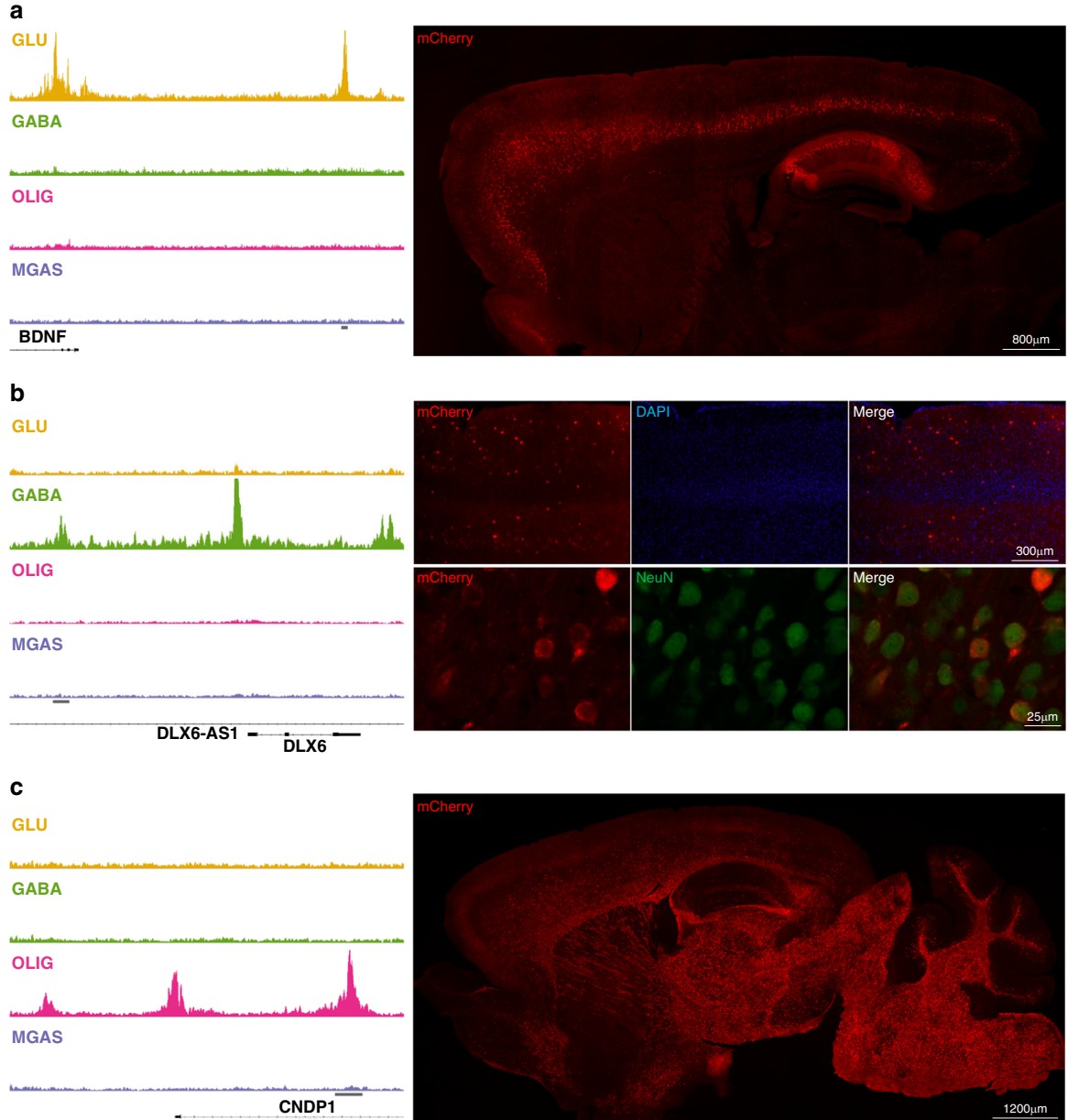

**Fig. 4 Transgenic evaluation of putative cell-type-specific enhancers.** Left column: Cell-type-specific OCRs identified by ATAC-seq and nearby genes: (**a**) glutamatergic (*BDNF*), (**b**) GABAergic (*DLX6*), and (**c**) oligodendrocytes (*CNDP1*). The horizontal gray bars denote OCR assayed in directed transcription via transgenesis. Right: Representative immunofluorescent images showing mCherry (red) expression in 30 μm thick sagittal sections from (**a**) *BDNF* and (**c**) *CNPD1* transgenic mice. In (**a**), specific mCherry expression is identified in Layer V of the cortex and in hippocampus. In (**b**) representative images of mCherry (red) staining in the cortex of *DLX6* transgenic mice (top panel) and double labeling with NeuN (green; bottom panel), showing expression restricted to neurons and scattered in the cortex, similar to the distribution of GABAergic interneurons. In (**c**) mCherry expression is shown to be restricted to white matter. Four image frames of three independent brain slices per each mouse were analyzed (BDNF enhancer *n* = 5; DLX6 enhancer *n* = 6; CNDP1 enhancer *n* = 4).

common risk variation with cell-specific epigenome sequences has the potential to identify cell types and molecular mechanisms that are relevant for the etiopathogenesis of a given disease. We previously demonstrated a significant enrichment of neuronal OCRs with schizophrenia and no significant overlap with Alzheimer's disease[5]. Here, we improved the resolution of these results by identifying glutamatergic cortical neurons, followed by GABAergic interneurons as the cell types most relevant to the etiology of schizophrenia. This is consistent with a recent study that leveraged cellular taxonomy of the brain from single-cell RNA-sequencing and mapped schizophrenia risk loci to similar cortical cell types[37]. In addition, we identified a nominally

significant enrichment for microglia/astrocyte OCRs in Alzheimer's disease, which is consistent with a recent study implicating microglia enhancers in Alzheimer's disease[38].

The cell-specific map of chromatin accessibility generated in this study can be used to deconvolute bulk tissue ATAC-seq data from the human brain cortex. Estimated cell type composition in bulk tissue data explained overall ~10% of variation in chromatin accessibility. By correcting individual heterogeneity in cell type composition, we substantially increased power to perform differential chromatin accessibility analysis among schizophrenia cases and controls. In addition, by using cell-specific OCRs, we ranked cell types based on the fraction of OCRs showing a

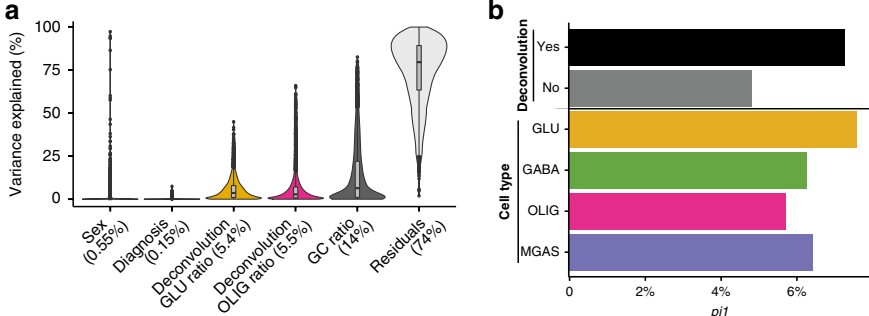

**Fig. 5 Overlap of OCRs with trait- and disease-associated genetic variants. a** *P*-value for enrichment of trait-associated genetic variants in cell-specific OCRs. The heritability coefficient of genetic variants overlapping different sets of OCRs in (**b**) SCZ and (**c**) AD. Positive coefficient signifies enrichment in heritability. "Multiregion" was our previous study of neuronal and non-Neuronal cells across multiple regions of the adult human brain. The region-specific OCRs are neuronal, as only neuronal cells showed a marked region variability. In all cases, the overlap was assessed using LD-score partitioned heritability where the OCRs were padded with 1000 bp to also capture adjacent genetic variants and corrected for the general genomic background. "#": Significant for enrichment in LD score regression after FDR correction of multiple testing across all tests in the plot (Benjamini & Hochberg); "·": Nominally significant for enrichment; DLPFC: dorsolateral prefrontal cortex. Error bars indicate standard errors from LD score regression using respectively 1,021,224 and 1,034,664 SNPs.

**Fig. 6 Using cell-specific OCRs for deconvolution of bulk ATAC-seq data. a** Violin plot illustrating the proportion of variation in chromatin accessibility explained by biological, technical, and cell-type deconvolution covariates. The GC ratio can be considered a signal to noise parameter. Numbers in parentheses indicate the median. **b** In various scenarios the proportion of non-null tests, *pi1*, was estimated for OCRs. Higher estimates indicate more significant differences between schizophrenia cases and controls. First, the effect of deconvolution on the *pi1* was assessed, and the addition of a deconvolution parameter was seen to increase power to detect case-control differences. Secondly, the pi1 estimates were calculated in just cell-specific OCRs. Here the largest case-control differences were seen in GLU-specific OCRs.

difference in chromatin accessibility between schizophrenia cases and controls. Overall, our analysis highlighted glutamatergic cortical neurons as the most perturbed cell type in schizophrenia based on two orthogonal approaches: (1) overlap with common genetic risk variation and (2) differential chromatin accessibility analysis. This provides additional support to the glutamate hypothesis of schizophrenia and is consistent with morphological alterations of dendrites of glutamatergic neurons in the cerebral cortex of individuals with schizophrenia[39].

Broader insight into gene regulation in the human brain could be gained by studying additional brain regions in additional cell types and at different developmental time points. It would be interesting to assay the epigenome of dopaminergic neurons and particularly their involvement in schizophrenia. We do, however, also note that our study encompassed the DLPFC, which has long been implicated in schizophrenia[40]. Our MGAS signal is a mixture of microglia and astrocytes, with the former being more prevalent[15], this likely biases towards this cell type, however, we did note a strong signal for both known microglia and astrocyte genes. Single-cell assays such as scATAC-seq provide unparalleled resolution to identify cell-specific OCRs; however, scATAC-seq data are very sparse with only one copy of each chromosome per cell. Therefore, the FANS-generated OCRs of this study will provide additional support to perform de novo taxonomy using single-cell data. Furthermore, chromatin accessibility could be studied in combination with additional epigenomic assays as well as gene expression, although the latter would suffer from the loss of cytoplasmic RNA when studying nuclei derived from frozen tissue specimens. Finally, future studies could examine OCR profiles of glutamatergic neurons in individuals with schizophrenia to elucidate which molecular mechanisms are affected.

In conclusion, the present study illustrates how studying open chromatin in different cell-types can be used to better understand gene regulation in the human brain and to interpret the impact of genetic variants associated with neuropsychiatric traits. We provide our results as a resource to further study gene regulation both genome-wide and at the single-gene level.

## Methods

**Description of the postmortem brain samples**. Brain tissue was obtained from four postmortem donors, three males and one female, ages 20–28 from a single brain collection. All four subjects (Supplementary Data 1) were of Caucasian ancestry based on self-report and genetic markers of ancestry. Samples were collected with a postmortem interval of less than 24 h at autopsy from The Department of Forensic and Insurance Medicine, Semmelweis University, Hungary, as previously described[41]. Informed consent from legally authorized representatives was obtained at the time of autopsy. The forensic pathologist performing the autopsy determined the cause of death. All were determined to have died a sudden, natural death. The subjects had no history of illicit substance abuse, alcohol abuse, or psychiatric disorders. The subjects had a negative toxicology and were not taking neuropsychiatric medications (including benzodiazepines, anticonvulsants, any antipsychotics, antidepressants, or lithium). To the best of our knowledge these sample characteristics should jointly preclude selection biases that would affect the postmortem brain analyses.

From each subject a dissection was obtained from the anterior cingulate cortex, the dorsolateral prefrontal cortex, and the primary visual cortex (Supplementary Fig. 1) yielding a total of 12 dissections. The Department of Forensic and Insurance Medicine, Semmelweis University provided ethical oversight. CerebroViz was used to illustrate the locations of these three brain regions[42].

**FANS sorting of four different cell types**. From each dissection, 50 mg of frozen brain tissue was homogenized in cold lysis buffer (0.32 M Sucrose, 5 mM CaCl$_2$, 3 mM Magnesium acetate, 0.1 mM, EDTA, 10 mM Tris-HCl, pH8, 1 mM DTT, 0.1% Triton X-100). Samples were then filtered through a 40 μm cell strainer, and the flow-through was underlaid with sucrose solution (1.8 M Sucrose, 3 mM Magnesium acetate, 1 mM DTT, 10 mM Tris-HCl, pH8) and subjected to ultracentrifugation at 107,000$g$ for 1 h at 4 °C. Pellets were resuspended in 500 μl DPBS and incubated in BSA at a final concentration of 0.1% together with anti-NeuN antibody (1:1000, PE conjugated, Millipore Cat FCMAB317PE), anti-SOX6[7] and

anti-SOX10[8]. Following incubation in primary antibodies, samples were subjected to a second ultracentrifugation step prior to incubation in secondary antibodies[4].

Preceding FANS sorting, DAPI (Thermoscientific) was added to a final concentration of 1 μg/ml. GABAergic neurons (DAPI+NeuN+ SOX6+), Glutamatergic neurons (DAPI+NeuN+ SOX6−), oligodendrocytes (DAPI+NeuN− SOX10+) and microglia/astrocytes (DAPI+NeuN− SOX10−) nuclei were sorted into individual tubes (pre-coated with 5% BSA) using a FACSAria flow cytometer (BD Biosciences) equipped with FACSDiva Version 8.0.1 software. Hence, each of the 12 dissected tissue samples yielded four different cell types giving a total 48 samples.

**Generation of ATAC-seq libraries and sequencing**. ATAC-seq reactions were performed on the 48 samples using an established protocol[43] with minor modifications. In brief, 75,000 sorted nuclei were centrifuged at 500$g$ for 10 min at 4 °C. Pellets were resuspended in transposase reaction mix (25 μL 2× TD Buffer (Illumina Cat #FC-121-1030) 2.5 μL Tn5 Transposase (Illumina Cat #FC-121-1030) and 22.5 μL Nuclease Free H$_2$O) on ice. Reactions were incubated at 37 °C for 30 min and then purified using the MinElute Reaction Cleanup kit (Qiagen Cat #28204), eluting in 10 μL of buffer EB. Following purification, library fragments were amplified using the Nextera index kit (Illumina Cat #FC-121-1011) under the following cycling conditions: 72 °C for 5 min, 98 °C for 30 s, followed by thermocycling at 98 °C for 10 s, 63 °C for 30 s, and 72 °C for 1 min for a total of five cycles. To prevent saturation due to over-amplification, a 5 μl aliquot was then removed and subjected to qPCR for 20 cycles for calculation of the optimal number of cycles needed for the 45 μL reaction that remained. How many additional cycles were needed was determined by first plotting linear Rn vs. Cycle and secondly calculating the cycle number corresponding to a quarter of the maximum fluorescence intensity. Adding two to four cycles to this estimate was found to yield optimal libraries, as evidenced by analysis on Tapestation D5000 ScreenTapes (Agilent technologies Cat# 5067-5588). Following amplification, libraries were resolved on 2% agarose gels and fragments ranging in size from 100-1000 bp were excised and purified (Qiagen Minelute Gel Extraction Kit—Qiagen Cat#28604). Before sequencing, libraries were quantified with the Qubit dsDNA HS assay kit (Invitrogen Cat#Q32851) and using quantitative PCR (KAPA Biosystems Cat#KK4873). Fragment sizes were estimated using Tapestation D5000 ScreenTapes (Agilent technologies Cat# 5067-5588) and libraries were sequenced on Hi-Seq2500 (Illumina) obtaining 2×50 paired-end reads.

**Processing of data**. An outline of the pipeline for the data processing is provided in Supplementary Fig. 2 and detailed in the following. R was used for job submissions and statistical analyses unless otherwise noted.

**Alignment of reads**. Sequenced reads were delivered by the sequencing facility already demuxed and with adaptors trimmed. FASTQ files were matched to the respective samples based on pooling IDs and barcodes. The one to one match and file integrity was confirmed using MD5 checksums.

Reads were subsequently aligned to the hg19 reference genome with the pseudoautosomal region masked on chromosome Y with the STAR aligner[44] (v2.5.0) and the following parameters:

    --alignIntronMax 1
    --outFilterMismatchNmax 100
    --alignEndsType EndToEnd
    --outFilterScoreMinOverLread 0.3
    --outFilterMatchNminOverLread 0.3.

For STAR and other java programs, java v1.7.0 was used. The alignment yielded a BAM file for each sample consisting of mapped paired-end reads sorted by genomic coordinates. From these files, reads that mapped to multiple loci were removed using samtools[45], duplicated reads were removed with PICARD (v2.2.4; http://broadinstitute.github.io/picard), and, finally, reads mapping to the mitochondrial genome were removed. On this file, quality was assessed using Qualimap, fastqc, and phantompeakqualtools. For the latter and other programs in python v2.7.14 was used.

**Genotype calling**. Genotypes were called using GATK (v3.5.0)[46], and the resultant files were compressed using bamUtil. In brief, the following steps were performed: (1) indel-realignment; (2) base score recalibration; and (3) joint genotype calling across all samples for variants with a phred-scaled confidence threshold > = 10. All clustered variants, variants in ENCODE blacklisted regions of the genome[47], and variants not in dbSNP v146[48] were not considered. Read depth was not used as a filtering criterion. Finally, only variants with minor allele frequencies (MAF) ≥ 25% were retained. For genotype file processing, bcftools, vcftools and plink was used. The genotype concordance amongst samples was quantified using both the fraction of concordant genotype calls and the kinship coefficient from KING v1.9[49]. Both approaches gave comparable outcomes, and both indicated an unambiguous separation of samples from the four different individuals.

**Peak calling and read quantification for quality control**. Peak calling was done using MACS v2.1 as previously described[5]. In short, we merged samples from the same brain region and cell type into one BAM-file (e.g. one file would contain all reads from the four samples of glutamatergic neurons derived from the ACC

dissections). The resultant twelve bam files were then subsampled to a uniform depth and used as an input for peak calling.

**Gender determination of samples.** Three different metrics were used to assess the gender of the samples: (1) The rate of heterozygotic genotyping calls on the X chromosome outside the pseudoautosomal regions. For this, variants with MAF < 5% were discarded. In samples from male individuals, a high heterozygosity rate potentially indicates sample contamination or an incorrect gender. (2) The read counts of OCRs adjacent to *FIRRE* and *XIST*, which predominantly show chromatin accessibility in samples from female individuals[50]. (3) Read counts in OCRs identified on the Y chromosome outside the pseudoautosomal region. No gender mismatches were identified using these three metrics.

**Metrics used for quality control.** For each sample, the following quality control metrics were used: the total number of initial reads; the number of uniquely mapped reads; the fraction of reads that were uniquely mapped; further metrics from the STAR aligner; the duplication and insert metrics from Picard; the rate of reads mapping to the mitochondrial genome; the PCR bottleneck coefficient (PBC), which is an approximate measure of library complexity estimated as uniquely mapped non-redundant reads divided by the number uniquely mapped reads; the normalized strand cross-correlation coefficient (NSC) and the relative strand cross-correlation coefficient (RSC), which are metrics that use cross-correlation of stranded read density profiles to assess sample quality independently of peak calling; and, finally, the fraction of reads in peaks (FRiP), which is the fraction of reads that fall in detected peaks, the fraction of reads in only blacklisted peaks, and the ratio between these two metrics (for these metrics the consensus set of peaks was used). The main quality metrics are shown in Supplementary Data 1.

**Quality control and further processing.** More than 28.9 million sequenced paired-end reads were obtained for each sample. Because of using FANS sorted nuclei as opposed to whole cells, only a low fraction of the reads mapped to the mitochondrial genome (mean 1.32% of the uniquely mapped reads). We examined libraries that had a low FRiP (<5%), had a low final read count (<5 million reads), visually were outliers in clustering, or looked to have outright failed when inspecting the bigWig track. In this analysis, one sample of GABAergic neurons from the anterior cingulate cortex was clearly of inferior quality and was left out, thereby leaving 47 samples for downstream analyses (Supplementary Data 1).

To assess the reproducibility among biological samples within the same cell type and brain region, we performed pairwise correlation between the raw read counts over consecutive bins of 10,000 bp genomic regions using bamCorrelate[51]. We also calculated transcription start site (TSS) enrichment in housekeeping genes as implemented in ataqv[52] for the current dataset as well as three other postmortem brain studies[5,31,53]. Ataqv calculates coverage around the TSS using ATAC-seq fragments up to 1 kb from the TSS in both directions.

The samples from the same brain region and cell type were subsequently subsampled and merged, creating 12 BAM-files (GLU from the anterior cingulate cortex, GLU from the dorsolateral prefrontal cortex, etc.) with a uniform depth of 98.1 million paired-end reads. Using these BAM-files, bigWig files were created using bedtools, bedGraphToBigWig, wigToBigWig, and wiggleTools. Peaks were called with the same parameters as for QC. A consensus set of peaks was subsequently created requiring a peak to be called in one or more of the merged BAM-files. After removing peaks overlapping blacklisted genomic regions, 177,178 peaks remained. Next, read counts of the individual 47 non-merged samples within these peaks were quantified, again, using the same parameters as for quality control.

To assess the potential functional impact of the identified OCRs, we evaluated the overlap with genetic variants affecting gene expression. In particular, we downloaded the variants annotated to the 95% credible set interval by Hormozdiari et al. for eGenes in GTEx brain samples[9,10], and used the genome-wide profile as background for evaluating whether variants in the ATAC-seq OCRs have a larger chance to affect gene expression. Significance was evaluated with a Fisher exact test. To rule out potential bias, we also permuted the positions of the ATAC-seq OCRs with consistent results. To assess the impact of adjacent regions, we also reanalyzed the overlap using various paddings around the OCRs.

**Analysis of cell and region-specific chromatin accessibility.** To assess which OCRs showed cell or regional specificity we performed differential chromatin accessibility analysis. For this, chromatin accessibility was estimated by the number of ATAC-seq reads overlapping a given OCR. The more overlaps seen with an OCR, the more accessible that OCR was deemed to be. The statistical analyses of these read counts were conducted as follows:

*First Read counts and OCRs were filtered.* A sample by OCR matrix of read counts was generated as described in the preceding section (47 samples by 177,178 OCRs). From this matrix, we excluded OCRs that were lowly accessible by only keeping OCRs that had at least 1 count per million reads in at least 10% of the samples. This removed just 26 OCRs and resulted in a final read count matrix of 47 samples by 177,152 OCRs.

Next, the read counts were normalized using the trimmed mean of M-values (TMM) method[54], but here only OCRs directly overlapping an autosomal

transcription start site (TSS) of a protein-coding gene were considered for calculating the normalization factor. This approach was employed based on the observations that the coverage of open chromatin and composition (promoter and non-promoter OCRs) varied by cell type. Non-promoter OCRs have on average less accessible chromatin and, hence, lower read counts. As TMM normalization includes a log transformation of read counts, it puts comparatively more emphasis on these non-promoter OCRs compared to if no transformation was used. Likely due to the two observed differences between cell types, we found standard TMM normalization to yield biologically implausible results with OCRs overlapping TSSs of housekeeping genes skewed towards being more accessible in one cell type than another when using a standard TMM normalization. When we used only the aforementioned promoter OCRs, we observed a balanced accessibility of housekeeping genes. We further evaluated the normalization procedure as follows: TMM normalization is always between pairs of samples, so one is chosen as a reference. In the edgeR implementation of TMM normalization, the sample whose upper quartile most closely matches the mean upper quartile is selected as the reference sample, which might not be equally applicable to ATAC-seq data on markedly different cell types. To examine this, we calculated the normalization factor, considering each sample in turn as the reference. This demonstrated that the choice of reference sample had only a very modest effect on the normalization factor (standard deviation of 0.0161, on average). However, we opted to use the mean of these normalization factors to avoid arbitrariness relating to which sample was used as the normalization reference.

As the next step, covariate exploration and model selection was carried out. Several biological and technical sample-level metrics were examined for their effect on the observed read counts. Here, some parameters were found to show different distributions in different cell types (e.g. yield of nuclei from FANS, the number of peaks called in a sample, FRiP, chrM metrics, RSC and NSC, and Picard insert metrics). We normalized such metrics to the median of each cell type.

To explore the effect of these technical and biological covariates, we first did a principal component analysis (PCA) on the normalized read counts to identify high-variance components explaining more than 1% of the variance. This was done separately for the different cell types as the PCA otherwise largely picked up the cell-type differences. We then accessed the correlation of covariates with the PCs and selected those that showed a significant correlation with one or more PCs at a lenient FDR cut-off of 0.2 as candidate covariates for the analysis of differential chromatin accessibility. This encompassed 39 covariates, including FRiP, mapping metrics, insert metrics, the rate of reads mapping to the mitochondrial genome, PBC, RSC, and barcode. These covariates were subsequently assessed as detailed in the following.

Next, the starting point for modeling chromatin accessibility was chosen with the variables "cell type by brain region" ($4 \times 3 = 12$ levels) and "gender" (2 levels) for a base model. "gender" was included as it is known to have a strong effect on a few OCRs primarily located on the sex chromosomes. To assess which covariates should be included in order to have a good average model for OCR accessibility, we employed the Bayesian information criterion (BIC). In particular, it was for each additional covariate tested how many OCRs showed an improved BIC score minus how many showed a worse BIC score when the covariate was included in the linear regression model compared to when it was not. Here, a covariate was required to net improve least 5% of the OCRs showed a change of 2 in the BIC score, which corresponds to the lower boundary of "positive" evidence against the null hypothesis[55], in order for it to be included in the final model. Initially, 34 numeric covariates were evaluated in this way. Compared to the base model, FRiP showed the largest and a very pronounced improvement in the fit of the model as it improved a net of 54.4% of the OCRs. After this variable was added to the base model and testing the remaining covariates against this new base model, no additional covariates were found to fulfill the criteria for inclusion.

Subsequently, 5 categorical covariates were considered for inclusion due to the higher number of degrees of freedom of each covariate. None of these fulfilled the BIC criteria for inclusion.

Finally, it was considered if the selected numeric covariate (FRiP) affected chromatin accessibility as a quadric term by testing the squared FRiP for inclusion. It did not meet the BIC inclusion criteria and was therefore not added to the model.

Resultantly the final model of chromatin accessibility included three variables: cell type by brain region (12 levels), gender (2 levels), and FRiP (numeric). This model jointly encompassed 14 degrees of freedom.

Following model selection, statistical analyses of differences in chromatin accessibility were carried out. To model the normalized read counts the *voomWithQualityWeights* function from the *limma* package[56] was used. This function employs both observational-level and sample-level weights. In particular, *voom* first residualizes the read counts and fits a mean-variance function across all OCRs to account for the fact that more accessible OCRs (e.g. those with higher log counts per million) show lower variance. The observation level weights are then set as the inverse of the estimated variance. Secondly, the sample weights are similarly estimated and used to calculate a final set of weights. For this, quantile normalization was not employed as it was found unfit to handle cells with different chromatin compositions. In particular, the different cell types had different proportions of promoter and non-promoter OCRs. They also had different compositions of highly accessible OCRs (primarily promoters) and lowly accessible OCRs (primarily non-promoters). As the quantile normalization forces all samples

to have the same empirical distribution function of read counts this would incur artefactual changes to the chromatin accessibility.

The normalized read count matrix from *voomWithQualityWeights* was then modeled by fitting weighted least-squares linear regression models estimating the effect of the right-hand side variables on the accessibility of each OCR: *chromatin accessibility ~ cell type:brain region + gender + FRiP*. In so doing, we model both cell type and brain region effects. In this model, the effect on the chromatin accessibility of an OCR can then be assessed by testing the coefficient of interest for being non-vanishing using the linear regression utilities implemented in *limma*.

As an initial assessment of how dissimilar the brain regions and cell types were, all pairwise statistical comparisons amongst the cell type by brain region groups were conducted (12 groups, 66 comparisons) giving in each comparison $P$-values for all OCRs. These $P$-values were then used to assess the proportions of true tests by estimating $pi1$ using "propTrueNull" function from limma with $pi1$ given as $pi1 = 1-pi0$. These measures were then used as a measure for how different the chromatin accessibility was in relevant comparisons such as between different cell types and brain regions.

To establish cell specificity of chromatin accessibility we first conducted all pairwise comparisons of chromatin accessibility between the cell types (e.g. GLU vs GABA). In each of these contrasts, the cells were compared with the respective cells of the same brain region, and significance was established as this contrast differing from 0 (e.g. p(GLU_ACC- GABA_ACC + GLU_DLPFC−GABA_DLPFC + GLU_PVC−GABA_PVC! = 0)). In this way, potential overall differences in the brain region are accounted for. We subsequently defined OCRs specific to the four different cell types (GLU, GABA, OLIG, and MGAS) by requiring an OCR to be significantly more accessible at FDR 5% in all pairwise comparisons. For example, the GLU-specific OCRs were defined as

GLU specific OCRs

= OCRs significantly more accessible in GLU than GABA

∩ OCRs significantly more accessible in GLU than OLIG

∩ OCRs significantly more accessible in GLU than MGAS

Consequently, such cell-specific OCRs are truly specific to the cell type in question. For example, an OCR showing high and comparable chromatin accessibility in GLU and GABA, and a low accessibility in OLIG and MGAS, would *not* be assigned as specific to GLU. The OCRs specific to neurons were defined as the union of GLU OCRs more accessible than either non-neuronal sample and GABA OCRs more accessible than either non-neuronal sample. The non-neuronal samples were similarly defined. Stated as sets this would be

Neuron specific OCRs

= (OCRs significantly more accessible in GLU than OLIG

∩ OCRs significantly more accessible in GLU than MGAS)

∪ (OCRs significantly more accessible in GABA than OLIG

∩ OCRs significantly more accessible in GABA than MGAS)

nonNeuron specific OCRs

= (OCRs significantly more accessible in OLIG than GLU

∩ OCRs significantly more accessible in OLIG than GABA)

∪ (OCRs significantly more accessible in MGAS than GLU

∩ OCRs significantly more accessible in MGAS than GABA)

Thus, these "Neuron" and "non-Neuron" sets also include OCRs that are specific to the overall cell group but might be equally accessible in both of the constituent cell subtypes. For instance, an OCR which is highly and equally accessible in GLU and GABA but lowly accessible in both OLIG and MGAS would be listed here as a "Neuronal" OCR. It would, however, not be listed as GLU or GABA specific, as detailed above. For the tables of these six sets of cell-specific OCRs we list the statistics for the least significant comparison. Next, to establish OCRs of a given cell type that showed regional differences, the linear model was fitted for the complete dataset, but to increase power in significance testing (e.g. topTable in *limma*) only consensus OCRs overlapping OCRs called in cell type in question were considered. For each cell type, one region was tested against the average of the other brain regions. For instance, GLU OCRs showing accessibility specific to ACC was assessed as OCRs having a higher accessibility in GLU ACC samples than the average of GLU DLPFC and GLU PVC samples. As with the cell-specific OCRs, an FDR cut-off of 0.05 was employed.

Using the cell-specific OCRs, we also compared the cell specificity identified here to that which was found in other human brain studies:

- ATAC-seq from Fullard et al. 2018 (115 NeuN+/− samples from 14 regions)[5]
- ATAC-seq from Rizzardi et al. 2019 (22 NeuN+/− samples from 2 regions)[53]
- RNA-seq from Rizzardi et al. 2019 (20 NeuN+/− samples from 2 regions)[53]
- RNA-seq from Mendizabal et al. 2019 (89 NeuN+/Olig2+ DLPFC samples)[57]

With the exception of Mendizabal et al., these studies did not provide the differential analysis results. We therefore reprocessed their raw data (GSE96613, GSE96614, GSE96949) using our pipeline, and ran a differential analysis. For Mendizabal et al., we used the differential expression analysis provided in their supplementary materials.

Finally, an adjusted matrix of chromatin accessibility was created where the effect of gender and FRiP was removed. This residualization was done by subtracting the estimated effect of these variables on the read count matrix and hence retaining just the effect of the cell type and brain region. Subsequently, this matrix was used as an input to Rtsne 0.13 with a perplexity parameter of 7 to do a t-SNE based clustering of the samples.

**Annotating OCRs.** The Ensembl 75 genes were used for all analyses in this paper. For the protein-coding genes, a few share the same gene name ($n = 81$). To have unique gene names, these were appended "_v1", "_v2", etc. Further, ChIPSeeker[58] was used to assign genomic context and the closest gene of the ATAC-seq OCRs. For ChIPSeeker, a transcript database was created using GenomicFeatures[39] and the Ensembl genes. Finally, the genomic contexts were defined as promoter (±3 kb of any TSS), 5′-UTR, 3′-UTR, exon, intron, and distal intergenic.

To compare the OCRs identified using ATAC-seq in this study to previously reported open chromatin and chromatin states from REMC[2,11], the overlap was calculated using the Jaccard index. The Jaccard index here was taken as the intersection of base pairs divided by union of base pairs. For the previously reported epigenome data, the imputed datasets were used due to their broader scope and higher quality[11]. From these datasets, the DNase-seq OCR sets and the chromHMM 25-state epigenome model were used. For the chromHMM model, related chromatin states were grouped as follows to reduce dimensions as and thus keep the analyses manageable: Promoter (Active TSS, Promoter Upstream TSS, Promoter Downstream TSS 1, Promoter Downstream TSS 2); and Primary enhancers (Active Enhancer 1, Active Enhancer 2, Active Enhancer Flank). We additionally organized the 127 epigenomic samples into four categories:

1. "Brain tissue": Brain Hippocampus Middle, Brain Substantia Nigra, Brain Anterior Caudate, Brain Cingulate Gyrus, Brain Inferior Temporal Lobe, Brain Angular Gyrus, Brain Dorsolateral Prefrontal Cortex, Brain Germinal Matrix, Fetal Brain Female, and Fetal Brain Male.

2. "Brain-derived cells": NH-A Astrocytes Primary Cells, Ganglion Eminence derived primary cultured neurospheres, Cortex-derived primary cultured neurospheres, H1 Derived Neuronal Progenitor Cultured Cells, H9 Derived Neuronal Progenitor Cultured Cells, and H9 Derived Neuron Cultured Cells.

3. "Immune": Primary mononuclear cells from peripheral blood, Primary T cells from peripheral blood, Primary T cells effector/memory enriched from peripheral blood, Primary T cells from cord blood, Primary T regulatory cells from peripheral blood, Primary T helper cells from peripheral blood, Primary T helper naive cells from peripheral blood, Primary T helper cells PMA-I stimulated, Primary T helper 17 cells PMA-I stimulated, Primary T helper memory cells from peripheral blood 1, Primary T helper memory cells from peripheral blood 2, Primary T CD8+ memory cells from peripheral blood, Primary T helper naive cells from peripheral blood, Primary T CD8+ naive cells from peripheral blood, Dnd41 TCell Leukemia Cell Line, GM12878 Lymphoblastoid Cells, K562 Leukemia Cells, Monocytes-CD14+ RO01746 Primary Cells, Primary monocytes from peripheral blood, Primary B cells from cord blood, Primary hematopoietic stem cells, Primary hematopoietic stem cells G-CSF-mobilized Male, Primary hematopoietic stem cells G-CSF-mobilized Female, Primary hematopoietic stem cells short term culture, Primary B cells from peripheral blood, Primary Natural Killer cells from peripheral blood, Primary neutrophils from peripheral blood, Spleen, Thymus, and Fetal Thymus.

4. "Other": all other REMC samples.

**Gene set enrichment analyses of open chromatin regions.** Gene set enrichment analyses were conducted as previously described[5]. In short, the number of OCRs overlapping the presumed regulatory domains of genes in a particular gene set is compared to OCRs overlapping any regulatory domains. Enrichment is then the OCR density for the regulatory domains of the gene set compared to the OCR density in the union of all regulatory domains.

**Direct mapping of open chromatin regions by overlap with TSS.** As a means to link OCR to gene(s) they are likely to affect, we intersected the OCRs with TSSs of Ensembl protein-coding genes, long noncoding RNA (lncRNA)[18], and micro-RNA[17]. For the protein-coding genes we only considered TSSs of protein-coding transcripts. For lncRNA we used the "Robust" assembly. For microRNA we only considered genes encoding a conserved microRNA according to TargetScan 7.1[59]. To estimate the specificity of the chromatin accessibility, we first used the resi-dualized read counts matrix described above and averaged across the four cell types and thus giving an OCR by four cell-type matrix. We next counted for each cell type and each gene the sum of ATAC-seq reads in any OCR overlapping the given gene's TSS(s). Together this yielded a gene by four cell types matrix. Using these counts we further calculated the fraction of reads originating from each cell type by dividing by the sum of reads for a gene. In the majority of cases at most one OCR overlapped the TSS(s) of a gene, and genes having no TSS overlapping any OCR were not considered. These counts were subsequently mapped back to the constituent OCRs and intersected with the cell-specific OCRs to obtain a direct

mapping of OCRs along with the fraction of reads in the gene's TSS(s) in each cell type as a measure of specificity.

**Overlap with common genetic variants**. To investigate if the OCRs of the brain cell types played a role in various diseases and traits, it was investigated if the OCRs were enriched in common trait-associated genetic variants identified using different GWAS studies. To do this, LD-score partitioned heritability[30] was employed. In LD-score partitioned heritability, it is calculated if common genetic variants in genomic regions of interest explain more of the heritability than variants not in the regions of interest, adjusting for the number of variants in either category. The approach allows for a correction of the general genetic context of the genetic regions of interest by using a baseline model of general genomic annotation (such as conserved regions and coding regions) and hence makes it possible to assess the enrichment above what is expected from the general genetic context of the genomic regions of interest. In addition to a *P*-value for this regression, a coefficient is outputted from the algorithm; to make this comparable across traits with a wide range of estimated heritability, we normalized it by the per-SNP heritability and called this the "heritability coefficient". This is different from the "enrichment" also outputted by the program in that the heritability coefficient takes the aforementioned baseline into account. We included the provided baseline model in our analyses and used the approach on a selection of neuropsychiatric[34,60–76] and unrelated[77–81] GWAS traits. In particular, one GWAS was tested at a time against the combination of one of our open chromatin datasets and the baseline model. When available the European only version of the summary statistics was used and, as a consequence, all GWAS results were European with the exception of coronary artery disease (which was 77% Europeans). The depression GWAS included only the subset of individuals in the publicly available summary statistics. The broad MHC-region (chr6:25-35MB) was excluded due to its extensive and complex LD structure, but otherwise default parameters were used for the algorithm.

**Predicting transcription factor binding using footprinting**. Using so-called footprinting analyses, ATAC-seq studies as well as similar approaches to study chromatin accessibility can be used for predicting transcription factor binding[20]. To do such footprinting analyses using the current dataset, a footprinting algorithm called PIQ[20] was applied to the 12 merged BAM-files. Motif selection, binding site filtering and estimation of gene regulation from TF binding was done as previously described[5]. In brief, transcription factor motifs were obtained from the CIS-BP database[21]. When multiple motifs were present for the same motif, the best one was selected in a majority vote-approach based on similarities between the motifs. Overlapping binding sites from the same transcription factor were discarded (e.g. palindromic motifs on opposite strands) preferentially keeping the binding site with the highest binding probability. When assigning binding sites to genes, the probability of the transcription factor regulating a given gene was approximated as an exponential decaying function. When a burden of "regulation" was estimated on the gene level, a non-redundant set of motifs were selected by iteratively removing motifs until none overlapped by 50% or more with 50% or more positional overlap. This was done, as transcription factors share motifs and without pruning, it would thus incur a bias.

Training of the PIQ model was done per cell type and subsequently applied to the three individual brain region sets of each cell type. To identify cell-type-specific regulatory differences at the gene level amongst the samples, three different sets of genes were used:

1. Ensembl protein-coding genes.
2. Long intergenic noncoding RNA (lincRNA) genes from a meta-assembly of lncRNA generated with the use of capped-end analysis of gene expression[18]. Of the different assemblies provided by the consortium, the default. "Robust" assembly of lincRNA genes was used. As a means primarily to focus on lincRNA genes transcribed independently of neighboring genes and to focus on lincRNA genes located in regions where it would be easier to identify differences in regulation unrelated to adjacent protein-coding genes, only lincRNA genes in the category "far from protein-coding genes" were considered. Further, lincRNA genes with coding status "uncertain" were excluded.
3. microRNA genes taken from miRBase 20[82]. Since some microRNA genes are believed to be erroneous annotations and/or non-functional, the microRNA genes that did not encode a conserved microRNA according to TargetScan 7.1 were discarded[59].

We next wanted to use the estimates of the gene level regulatory burden to access how the genes were regulated differently in the different cell types. For this we did pairwise comparisons of samples in each case calculating the difference in regulatory burden between the two samples and used these to rank the samples (from $-\frac{n-1}{2}$ to $\frac{n-1}{2}$). We then averaged such pairwise ranks amongst samples of interest. To establish a neuronal versus non-neuronal samples we took the mean of all the pairwise comparisons: on one side a neuronal sample and on the other side a non-neuronal sample. Similarly, for each of the four cell types, we took the mean of the rankings from all the pairwise comparisons: on one side a sample from the cell of interest and on the other a sample not from that cell type. To avoid incorrectly

counting the regulatory burden of a TF binding site on a gene multiple times, only non-redundant TF motifs were considered in all of the pairwise comparisons.

**Transcription Factors showing cell specificity**. In an attempt to identify TFs that were important to the different cell types, it was examined which TF motifs were overrepresented in the cell-specific OCRs. For this, we first created a consensus list of predicted binding sites from the union of TF binding sites that were considered to be bound in one or more samples based on the previously outlined criteria. This consensus list was then used to identify overrepresentation in the cell-specific OCRs. In particular, the fold enrichment was calculated and the statistical significance established by a one-sided binomial test accounting for the coverage of the OCRs. For the GLU, GABA, OLIG, and MGAS samples the background was for a given cell type taken as the union of OCRs specific to the other cell types. For neuronal vs. non-neuronal comparison the background was taken as the OCRs specific to the opposite cell type.

For neuronal samples vs. non-neuronal samples we aggregated all comparisons of, on one side, neuronal samples, and the other, non-neuronal samples. For each of the four cell types, we aggregated all comparisons of samples for the given cell-type on one side and samples of the remaining cell-types on the other, between samples from different cell types and brain regions. These scores were subsequently aggregated.

**Deconvoluting homogenate ATAC-seq samples in schizophrenia**. A study of ATAC-seq homogenate brains of individuals with and without schizophrenia was used to quantify the effect of dissection and cell composition bias on the identification of disease-related epigenetic changes[31]. 41 out of 288 samples were discarded as they had unrelated phenotypes and/or possible contamination[83]. Subsequently, a similar approach as applied to the ATAC-seq samples of this paper was used to process the data and identify covariates for differential analysis between schizophrenia cases and controls. Compared to the base model, i.e., *Diagnosis + Sex* (2 ×2 levels), *mean GC content per sample* showed the largest and very pronounced improvement in the fit of the model as it improved a net of 60.2% of the OCRs (i.e., 58,824 out of 97,688. OCRs). In the next rounds of BIC, the model selected also the ratio of glutamatergic neurons and ratio of oligodendrocytes that improved a net of 20.1% and 5.0% of OCRs, respectively. No other covariate fulfilling the criteria for inclusion was found.

Next, a reference panel of marker OCRs that capture cell type differences was derived from differential analysis results among cell-type-specific samples. We required marker OCRs to be cell-type-specific when compared against all other cell types at significance threshold of FDR < 5% and log(fold change) >2. Using these criteria, the following numbers of marker OCRs were obtained: 685 (glutamatergic neurons), 2,259 (GABAergic neurons), 890 (oligodendrocytes) and 1,511 (astrocytes and microglia). Then, for all ATAC-seq homogenate samples (see previous section), we quantified their accessibility in marker OCRs and used such epigenetics profiles as an input of dTangle[84], a deconvolution method built on the linear mixing model of linear-scale expressions of known marker OCRs.

**Mice**. Experimental procedures were carried out in compliance with the United States Public Health Service's Policy on Humane Care and Use of Experimental Animals and were approved by the Institutional Animal Care and Use Committee at Icahn School of Medicine at Mount Sinai. We used an approved protocol with the number16-0847. The IACUC is LA09-0272.. Both male and female B6D2F1/J transgenic founders were used for analysis. 6-week-old male and female B6D2 F1 hybrid mice were used as the basis for transgenic strains. Mice were maintained at 22±2 °C and 40-60% humidity with 12 h light/dark cycle, with ad libitum access to food and water.

**Vector**. The vector is shown in Supplementary Fig. 14, and contained the minimal hsp68 promoter, a heterologous intron, mCherry as reporter and 5' and 3' insulators to minimize effects of transgene insertion sites. CTCF insulator sequences were taken from litterature[85]. The transgenic vector was constructed by GeneBridges (Heidelberg, Germany). Cloning strategy is available upon request. The following human enhancer candidates were assayed with 50 bp padding (hg19):

Glutamatergic (BDNF): chr11:27782533-27783536
GABAergic (DLX6): chr7:96626530-96627270
Oligodendrocytes (CNDP1): chr18:72208398-72209567
Microglia/astrocytes (TYROBP): chr19:36400061-36400725

**Transgenesis**. The generation of transgenic mice was performed as previously described using standard pronuclear injection protocols[86] at ISMSS Mouse Genetics and Gene Targeting core, directed by Dr. Kevin Kelley. Transgenic founders were identified by PCR analysis of genomic tail DNA (Extract-N-Amp™ Tissue PCR Kit; Sigma, St Louis, MO) with the following mCherry specific primers:

Forward GGAGGATAACATGGCCATCATCAAGG
Reverse CGTACTGTTCCACGATGGTGTAGTCCTCG

**Histology**. Adult mice at 6 weeks of age were deeply anesthetized with pentobarbital (50 mg/kg), and intracardially perfused with ice-cold PBS followed by 4% paraformaldehyde in PBS. Brains were removed and post-fixed by overnight immersion in the same solution and then stored in PBS at 4 °C. Serial sagittal sections were obtained on a Leica microtome at 30 μm. All the following steps were done with gentle shaking. Free-floating brain sections were washed two consecutive times in 1× PBS, 10 min each, permeabilized in PBS with 0.25% Triton X-100 and 5% goat serum for 1 h at room temperature. Free-floating sagittal sections were incubated with the corresponding primary antibodies in PBS with 0.05% Triton X-100 and 1% goat serum: rat anti-mCherry (1:10000; Invitrogen; M11217), rabbit anti-NeuN (1:1000; Millipore; ABN78), rabbit anti-GABA (1:2000; Sigma; A2052), rabbit anti-PARV (1:1000; Swant; PV27), rabbit anti-Somatostatin (1:500; Peninsula; T-4103), rabbit anti-NPY (1:500; Abcam; ab30914), rabbit anti-VIP (1:1000; Immunostar; 20077), rabbit anti-Iba1 (1:500; WAKO; 019-19741), rabbit anti-Olig2 (1:500; Abcam; ab136253), rabbit anti-Ctip2 (aka Bcl11b; 1:500; Bethyl Laboratories; A300-385A), and rabbit anti-GFAP (1:500; DAKO; Z0334). Sections were washed three consecutive times, 10 min each, with 1× PBS plus 0.1% Triton X-100, then incubated for 2 h at room temperature with the appropriate secondary antibody: anti-rat Alexa Fluor 594 (1:500; ThermoFisher; A11007) and anti-rabbit Alexa Fluor 488 (1:500; ThermoFisher; A11034). Nuclei were visualized by DAPI staining (1:10000; ThermoFisher; 62248). Negative controls were performed for each primary antibody. Slides were sealed with Vectashield hard-set mounting medium (H-1400, Vector Laboratories) No signal was detected in sections incubated in the absence of the primary antibody. Layer V of the cortex was identified by comparison of mCherry(+) sections to sections stained with 0.1% cresyl violet.

Images were acquired with a Zeiss 700 confocal microscope (Zeiss, Thornwood, USA). For co-localization experiments, we acquired four image frames of three independent brain slices per each mouse (BDNF enhancer $n = 5$; DLX6 enhancer $n = 6$; CNDP1 enhancer $n = 4$; TYROBP enhancer $n = 4$) using a ×40 objective. Images were analyzed using ImageJ v1.51.

Additional images were acquired with an Olympus BX61 fluorescence microscope and processed with ImageJ v1.51.

**Reporting summary**. Further information on experimental design is available in the Nature Research Reporting Summary linked to this paper.

## Data availability

The ATAC-seq data generated as part of this publication have been deposited in Gene Expression Omnibus and are accessible through GEO Series accession number "GSE143666". Further, UCSC tracks and downloads are provided at our webpage http://icahn.mssm.edu/boca2. The following reference datasets were downloaded from for the purpose of comparison with our study: https://www.synapse.org/#!Synapse:syn5584622, https://www.ncbi.nlm.nih.gov/geo/query/acc.cgi?acc=GSE96614, https://www.ncbi.nlm.nih.gov/geo/query/acc.cgi?token=ofwzsggybrihviv&acc=GSE96949, https://www.ncbi.nlm.nih.gov/geo/query/acc.cgi?acc=GSE108066. The following online resources and databases were used: dbSNP: https://www.ncbi.nlm.nih.gov/snp, REMC: http://www.roadmapepigenomics.org, MSigDB (GO, KEGG, and Biocarta gene sets): https://www.gsea-msigdb.org, lincRNA and microRNA from FANTOM: https://fantom.gsc.riken.jp, and miRBase: http://www.mirbase.org.

Summary statistics are available from the following links: "Complex Trait Genetics Lab [ctg.cncr.nl/software/summary_statistics]", "Coronary Artery Disease [cardiogramplusc4d.org]", "Genetic Investigation of ANthropometric Traits [portals.broadinstitute.org/collaboration/giant]", "International Inflammatory Bowel Disease Genetics Consortium [ibdgenetics.org]", "The Psychiatric Genomics Consortium [med.unc.edu/pgc]", "Social Science Genetic Association Consortium [thessgac.org/data]".

All other relevant data supporting the key findings of this study are available within the article and its Supplementary Information files or from the corresponding author upon reasonable request. A reporting summary for this article is available as a Supplementary Information file.

## Code availability

The analysis was done using free, publicly available software programs and libraries. In particular, the ATAC-seq reads were aligned with STAR [github.com/alexdobin/STAR], and analyzed using the R packages limma and edgeR [cran.r-project.org]. LDSc [github.com/bulik/ldsc] was used for integration with GWAS data. A stepwise approach of the deconvolution procedure of bulk ATAC-seq results is shown at http://icahn.mssm.edu/boca2.

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

## Acknowledgements

The authors thank the authors of the various GWASs for making summary statistics publicly available. Data on coronary artery disease was contributed by CARDIo-GRAMplusC4D investigators. This work was supported by the National Institutes of Health (R01AG050986 Roussos, R01MH109677 Roussos, U01MH116442 Roussos/Dracheva, R01MH110921 Roussos) and the Veterans Affairs (Merit grant BX002395 Roussos). This study was additionally funded by The Lundbeck Foundation, Denmark (Grant numbers R102-A9118 and R155-2014-1724). J.B. was supported in part by NARSAD Young Investigator Grant 27209 from the Brain & Behavior Research Foundation. Further, this work was supported in part through the computational resources and staff expertise provided by Scientific Computing at the Icahn School of Medicine at Mount Sinai and the assistance of members at the Mount Sinai Flow Cytometry CoRE. The funders had no role in the design and conduct of the study; collection, management, analysis, and interpretation of the data; preparation, review, or approval of the manuscript; and decision to submit the manuscript for publication.

## Author contributions

M.E.H., S.D., M.E.E., J.F.F., and P.R. contributed to experimental and study design as well as planning analytical strategies. Y.L.H. provided human brain tissue. A.K. and M.W. prepared the nuclei. J.F.F. conducted FANS and generated ATAC-seq libraries. J.C.M., S.C., C.C., and H.K. performed in vivo validation experiments. M.E.H., J.B., and P.R. carried out primary data analyses. B.Z. conducted eQTL overlap analyses. M.E.H., J.B., A.D.B., J.F.F., and P.R. wrote the manuscript.

## Competing interests

The authors declare no competing interests.
