## [Peer Review File · Nature Communications]

REVIEWER COMMENTS

Reviewer #1 (Remarks to the Author):

Identification of the specificity of open chromatin regions (OCRs) in different brain regions and cell types provides a novel avenue to understand the molecular diversity in the human brain. Furthermore, the integration of external databases and application of the cell type specific OCRs identified in this study as markers for computational deconvolution method provides strong evidence for the importance of cell type specific analysis. However, a major issue with this study is the low number of patients (n=4) and that there are already a number of published studies of isolated cells and single cells maps of brain cell types, which also looks at these cell types and in several instance with more resolution (i.e., PMID: 31042697, 30760929, 32139688 and data in PsychEncode). The novelty of this paper is that it offers more brain regions, specifically the cingulate and visual cortex, but the benefit of these additional regions for the schizophrenia work included in this paper is not clear. Another concern is that many of the analyses tells us what we already know about the different cell types investigated (i.e. there are cell type differences in open chromatin regions, GABA neurons have GABA receptor activation, glutamatergic neurons are linked to neuropsychiatric disease risk variants). Though such information gives the reader confidence in the quality of the cell material being analyzed, the paper overall is limited in terms of adding new biology to what we already know. Therefore, this paper though well written and carefully prepared, it has some important limitations.

Other Major Issues

1. The microglia and astrocyte signal (MGAS) is mixed, which likely signifies an astrocyte-dominant signal due to the predominance of this cell type
2. Please explain why almost half of all the OCRs identified in each cell type are unique for only a single replicate per group.
3. The authors mention in a comparison across brain regions that there are only statistical differences OCRs in glutamate neurons (line 148). However, with such a small number of patient samples, is such a statistical analysis meaningful? Does the identification of no significant difference in GABA, OLIG and MGAS, say anything biologically interesting?
4. Can the authors statistically test and explicitly mention in the results that greater number of OCRs found in glutamate neurons is not driven by sequencing read depth or genome coverage differences in glutamate neuron?
5. Comparisons with Roadmap DNase data in figure 2b states predominant overlap. Please provide an enrichment score to prove that this is not by chance. Without enrichment analysis, any interpretation of the overlap, such as MGAS enrichment in immune cells, is less reliable.
6. The enhancer construct analysis in specific cell types is a nice addition to the paper. However, the results are difficult to interpret. Why does only cortical layer 5 neurons express glutamate cell-specific enhancer (BDNF enhancer) and why is there a relatively low colocalization between the GABA cell-specific enhancer (DLX6), while the MGAS cell specific enhancer (TYROBP) is not observed in MGAS cells? Could there be species differences that affect the interpretability of the results? Would in vitro/human cell culture experiments have been a better choice? If enhancer constructs are a concern (as mentioned in the discussion) analysis of cell-type specific Hi-C data could be an option.
7. The use of LD score regression to examine the overlap of disease risk SNPs is nicely done. However, the diseases selected (neuropsychiatric diseases) are rather expected to be associated with neuronal OCRs based on several other publications (i.e. Price et al. 2019 Genome Biology, Polioudakis et al., 2019 Neuron). Could it be more interesting to examine overlap with the 170+ diseases in the LDscore regression tool with the brain cell types in this study? This could generate new leads on the contributions of these cell type to other diseases.

Minor Issues

1. The three brain regions studied should be mentioned in the abstract
2. The abstract states that glutamatergic neurons are the most affect cell type in schizophrenia –

this is a somewhat misleading statement because though true among the cells studied in this paper, there was no analysis of dopaminergic cell type which likely also have a strong role in this disease.

3. The first sentence of the second paragraph in the introduction need to be changed to clearly explain the limitation in the field.
4. For bioinformatic outlier removal, please provide a read or peak coverage and/or correlation value threshold.
5. The authors should mention the age and postmortem interval of the samples. Perhaps this could contribute to the 34% residual value in Fig 1f.
6. Promoter accessibility changes with both brain region and cell type. Was this taken into account during the overlapping analysis?
7. Why is the protein coding gene map comparison to GREAT in cell-specific pathways enrichments not provided as a supplementary figure (line 211)? Should a correlation analysis be included?
8. Figure 3b does not clearly show the overlap of TFs from multiple cell types. For example, the oligodendrocyte lines are over the non-neuronal lines making it difficult to see the overlap in TF like SF family. Separate cell type plot or modifying the existing plot could improve the clarity of this figure.
9. Multiple figures have axis too small to effectively read, making it difficult to interpret data. Please edit all figures accordingly.

Reviewer #2 (Remarks to the Author):

The study by Dr. Roussos and colleagues carries out ATAC-seq on 3 cerebral cortex regions in 4 individuals using FANS for NeuN, Sox6 and Sox10 to delineate 4 distinct cell types GABA neurons, Glu neurons, oligodendrocytes and microglia/astrocytes. This study follows up on the 2018 Genome Research paper of the same group which characterised NeuN+ and NeuN- cell populations, by further refining cell types.

The study is thorough, well written, and carries out standard analyses of ATAC-seq data, such as overlap analyses with epigenomic features, TSS, TF binding sites etc., as well as assessment of enhancer function in vivo for several open-chromatin regions.

1. The QC FRiP score are quite low: 0.05-0.15, while the standard for Encode data for example is >0.2. The authors need to provide additional information to demonstrate an acceptable signal-to-noise ratio, such as, for example a TSS enrichment score (<https://www.encodeproject.org/data-standards/terms/#enrichment>) and a replication rate or a correlation across replicates (i.e. different individuals , same brain region and cell type).
2. A comparison between the current ATAC-seq data and RNA-seq data from NeuN+/NeuN- cells (eg. PMID 31288836) is necessary for a functional interpretation of the open chromatin regions, rather than just overlaps (with gene sets from the Epigenomics Roadmap consortium).
3. The Jaccard index is not appropriate for calculating the overlap with different epigenomic features, as it does not take into account genome-wide background. A test with random sampling of the genomic background for each chromHMM category should be used instead.
4. For the variant enrichment analyses, an enrichment of eQTL data (eg. GTEX) would be valuable to demonstrate the enrichment of regulatory variants in the OCRs identified in the present study.
5. The title is an overstatement: there is no assessment of the function of glutamatergic neurons and no experimental evidence for genetic variants associated with schizophrenia affecting such function.

Minor:

FigS6,7: It's unclear what Log2Fc refers to, as these are enrichment analyses.

Reviewer #3 (Remarks to the Author):

This paper provides the first ATAC-Seq study to examine subtypes of neurons and non-neuronal cells. Therefore, although the sample size is small I believe it is a strong publication (indeed, it provides a further example of the power of within-subjects analyses of this type). The authors should be commended for making their data open access, and for providing it for review - this greatly enhances both the potential impact of the paper and the ease of review. The manuscript is generally well written and the figures and tables are largely clear. I therefore have only questions of clarification:

1. On p7, and in Fig 2b, I believe that the REMC comparisons would benefit from clarification. For example, the text states "[...] our 160 cell-specific OCRs predominantly overlap with REMC brain samples". Is this a typo? If not, please clarify how OCRs overlap with samples. Similarly, in Fig 2b, please state what the asterisks represent (and clarify why some bars have multiple asterisks).

2. Table 1 also requires clarification: Table 1: where were the neuronal and non-neuronal data in this table obtained from? If they are simply the Glu + GABA and OLIG + MGAS why do the values not add up?

3. Regarding the GWAS-related findings, were results corrected for between-phenotype correlations? Furthermore, it would be good to add comment about the diagnostic categories that *don't* show clear association, e.g. bipolar. Furthermore, what corrections (if any) were run for multiple comparisons for these studies?

4. It would be good to have more information about the postmortem brains included here – pH, and ideally RIN – I realise that this study did not examine RNA directly, but these measures still speak to overall tissue quality. It might be that some of this information is in Table S1, but the table is only partially readable as PDF (e.g. what was the cause of death of Female sample #4)

Minor point:

Why did the authors use Hg19 for their annotations? Could they provide a liftover to Hg38?

REVIEWER COMMENTS

Reviewer #1 (Remarks to the Author):

Identification of the specificity of open chromatin regions (OCRs) in different brain regions and cell types provides a novel avenue to understand the molecular diversity in the human brain. Furthermore, the integration of external databases and application of the cell type specific OCRs identified in this study as markers for computational deconvolution method provides strong evidence for the importance of cell type specific analysis. However, a major issue with this study is the low number of patients (n=4) and that there are already a number of published studies of isolated cells and single cells maps of brain cell types, which also looks at these cell types and in several instance with more resolution (i.e., PMID: 31042697, 30760929, 32139688 and data in PsychEncode). The novelty of this paper is that it offers more brain regions, specifically the cingulate and visual cortex, but the benefit of these additional regions for the schizophrenia work included in this paper is not clear. Another concern is that many of the analyses tells us what we already know about the different cell types investigated (i.e. there are cell type differences in open chromatin regions, GABA neurons have GABA receptor activation, glutamatergic neurons are linked to neuropsychiatric disease risk variants). Though such information gives the reader confidence in the quality of the cell material being analyzed, the paper overall is limited in terms of adding new biology to what we already know. Therefore, this paper though well written and carefully prepared, it has some important limitations.

Response:

The goal of this study is to generate a multi-regional atlas of chromatin accessibility, which extends our previous efforts (PMID: 29945882) by increasing the number of captured cell types. We would like to first acknowledge that there are multiple shortcomings of epigenomics/transcriptomics based on FANS/flow cytometry. Isolating human brain cells for studies is challenging and, although nuclear isolation from frozen specimens is possible this approach is only limited to broad defined cell types where available antibodies can be used to sort those nuclei. On the other hand, single cell approaches offer an agnostic, unbiased approach to capture the cellular heterogeneity. However, there are still limitations in single cell approaches, particularly when it comes to epigenomics as each nucleus contains only one copy of each chromosome. This means that the data is extremely sparse and due to its lower yields, often does not accurately quantify differences in chromatin accessibility. We briefly hint at this in the discussion, and have now elaborated slightly on this. Considering orthogonal datasets such as the chromatin accessibility map of the present study will improve signal to noise ratio and perform more robust de novo taxonomy in single cell data sets. The articles mentioned above are all single cell RNA-seq, and not ATAC-seq. Given that single nuclei ATACseq becomes the most popular epigenomic assay (and the one that is widely used in psychENCODE, AMP-AD and other consortia), it is critical to define robust markers for broad cell types in the human brain tissue that will be used as a reference map to define cell types with higher resolution. As for the PsychEncode data, the existing ATACseq data are primarily on bulk tissue or neuron vs non-Neuron FANS, though there are ongoing efforts to expand this to further cell types. Importantly, as noted by Reviewer #3: "This paper provides the first ATAC-Seq study to examine subtypes of neurons and non-neuronal cells."

We acknowledge that a lot of focus in the paper is on quality assurance and on confirming established brain biology, both necessary to substantiate the data set as a future resource for the scientific community. Amongst the novel findings in the paper, we would in addition to our comment above and the novelty indicated by the reviewer him-/herself like to highlight how we used the data to identify novel cell specific enhancers. Further, to the best of our knowledge, glutamatergic epigenome has not previously been directly linked to common schizophrenia risk variants.

As for the number of individuals in the study ($n=4$) with a total of twelve samples per cell type, this is not inconsistent with previous efforts such as Roadmap for epigenome, ENCODE and the reference brain atlas by psychENCODE. The various analysis in the paper as well as visual inspection of the individual bigWig tracks highlight the marked differences in chromatin accessibility between the cell types and the comparatively small differences between brain regions and individuals. The high number of open chromatin regions showing cell type differences in our quite strict approach (see also below) identified a high number of differentially accessible sites, again indicating that our study is well powered to study such cell type differences.

As for the regional differences, it would admittedly have been beneficial with a higher sample count. Here the contrasts for each cell type include twelve samples (e.g. GLU ACC samples versus the average of GLU DLPFC and GLU PVC samples), but only showed significant differences for glutamatergic neurons. This is of interest when interpreted together with the variance partitioning analyses and π_1 estimates. In the variance partitioning, the brain region added 3%-4% to the variance explained to the model (a point estimate) compared to the 1.5% of variance explained by the person, which in both cases is vastly lower than the variance explained by cell type alone (54%-58%). The π_1 estimates told a similar story. This leads us to conclude that "Chromatin accessibility was found to vary vastly by cell type and, more moderately, by brain region". A larger sample size would probably have afforded significant findings in all cell types. We've added a comment regarding this to the discussion.

Other Major Issues

1. The microglia and astrocyte signal (MGAS) is mixed, which likely signifies an astrocyte-dominant signal due to the predominance of this cell type

Response:

We thank the reviewer for pointing out that the relative signal coming from these subtypes and its importance haven't been stated in the paper. We have now added this to the discussion. Despite the ratio of the microglia vs astrocytes likely being skewed, we do in fact see both the astrocyte and microglia signatures as evidenced by visual inspection of chromatin accessibility around known marker genes as well as the overlap with single cell approaches as in supplementary figure 6a.

2. Please explain why almost half of all the OCRs identified in each cell type are unique for only a single replicate per group.

Response:

In our supplementary methods, we did not provide any information directly about the peak calling, but simply referenced a previous study. In light of this comment and in realizing the importance of this in interpreting the aforementioned figure, we've added this section to the supplement "In short, we merged samples from the same brain region and cell type into one BAM-file (e.g. one file would contain all reads from the four samples of glutamatergic neurons derived from the four ACC dissections). The resultant twelve bam files were then subsampled to a uniform depth and used as an input for the peak caller." This merging of files prior to peak calling was done to increase power for detecting open chromatin regions that were only accessible in a subset of cells or cell subtypes.

There are multiple potential explanations for a large part of the open chromatin regions being only identified in one out of three merged BAM-files from the same cell type. One could be that the OCRs show some/complete specificity to the brain regions assayed. Supporting this idea is the fact that GLU and GABA had the highest number of unique open chromatin regions and were also the two cell types showing the highest π_1 estimates

for brain region variability. Another contribution to the uniqueness of the identified cell types could be stochastic. Given a random sampling of reads and noise, a true peak might be called in one sample where the reads by chance resembles what the peak caller expects an open chromatin region to look like, whereas in another it might look more like random reads/noise. Open chromatin regions might simply not consistently show a sufficient signal if the region is only accessible in a small subset of cells and/or cell subtypes. Supporting this theory is the fact that open chromatin at promoters, which have comparatively higher accessibility than the other open chromatin regions, are more consistently called. A corollary of this would be that more open chromatin regions are yet to be discovered. Considering the uniqueness of the open chromatin regions in the cell types separately (e.g. the colored bars in figure 1c), which is relevant considering the massive differences in chromatin accessibility across cell types, 66%-80% of the promoter open chromatin regions were called in two or more brain regions. For non-promoter open chromatin regions these numbers ranged from 50%-67%.

3. The authors mention in a comparison across brain regions that there are only statistical differences OCRs in glutamate neurons (line 148). However, with such a small number of patient samples, is such a statistical analysis meaningful? Does the identification of no significant difference in GABA, OLIG and MGAS, say anything biologically interesting?

Response:

This comment has been addressed in the response to the reviewers introductory section

4. Can the authors statistically test and explicitly mention in the results that greater number of OCRs found in glutamate neurons is not driven by sequencing read depth or genome coverage differences in glutamate neuron?

Response:

As mentioned in the response to comment #2, we've now given an explicit statement about how the open chromatin regions were identified including a statement about the BAM-files used for the peak-calling being subsampled to uniform depth. We apologize for this not being specifically mentioned in the initial submission. The peak-caller, when applied to open chromatin assays such as ATAC-seq identifies regions where there are significantly more reads compared to the background. Given the subsampling and the internal mechanism of the peak-calling algorithm, the higher number of peaks cannot be down to sequencing depth/coverage differences.

5. Comparisons with Roadmap DNase data in figure 2b states predominant overlap. Please provide an enrichment score to prove that this is not by chance. Without enrichment analysis, any interpretation of the overlap, such as MGAS enrichment in immune cells, is less reliable.

Response:

The Figures 2b and S7 (previously called S5) were based on the overlap with the 127 different samples categorized into the four different types of samples ("Brain derived cells", "Brain tissue", "Immune" and "Other"). In this section we provide A) evidence that our statements about differences in overlap between the annotations are not down to chance. B) An analysis of enrichment against the genomic background.

To assess the significance of these differences between the cell specific OCRs we've now regressed the Jaccard overlap with these four roadmap sample categories against taking the "Other"-category as the intercept. This is not to evaluate the enrichment against the genomic background but the differences between the datasets. We've added this analysis to the text.

In this regression analysis, "Brain derived cells" and "Brain tissue" was significantly enriched compared to the "Other" Roadmap samples in all of our ATAC-seq peaks. For "Immune" there was at least a nominally significant depletion for "Glutamatergic", "GABAergic", and "oligodendrocytes", whereas "Microglia & Astrocytes" showed a nominally significant enrichment. The full results are shown below for the reviewers:

Glutamatergic

```
#####
                Estimate   Std. Error   t value   Pr(>|t|)
(Intercept)      0.026931026  0.0004710918  57.167253  1.893457e-90
myCats.4.catsBrain derived cells  0.008813518  0.0017938642   4.913146  2.794514e-06
myCats.4.catsBrain tissue      0.013135347  0.0014211053   9.243049  9.067937e-16
myCats.4.catsImmune            -0.007190621  0.0009061632  -7.935238  1.105181e-12
```

GABAergic

```
#####
                Estimate   Std. Error   t value   Pr(>|t|)
(Intercept)      0.022083958  0.0002762632  79.938100  6.940597e-108
myCats.4.catsBrain derived cells  0.004563740  0.0010519790   4.338242  2.961864e-05
myCats.4.catsBrain tissue      0.006025879  0.0008333814   7.230639  4.469606e-11
myCats.4.catsImmune            -0.004780566  0.0005314029  -8.996122  3.529452e-15
```

Oligodendrocyte

```
#####
                Estimate   Std. Error   t value   Pr(>|t|)
(Intercept)      0.040845041  0.0007386238  55.298838  9.690587e-89
myCats.4.catsBrain derived cells  0.007967411  0.0028125958   2.832761  5.394327e-03
myCats.4.catsBrain tissue      0.019247954  0.0022281479   8.638544  2.494574e-14
myCats.4.catsImmune            -0.003330996  0.0014207713  -2.344498  2.065513e-02
```

Microglia & Astrocytes

```
#####
                Estimate   Std. Error   t value   Pr(>|t|)
(Intercept)      0.046654595  0.0007340164  63.560701  6.286672e-96
myCats.4.catsBrain derived cells  0.009074932  0.0027950511   3.246786  1.504255e-03
myCats.4.catsBrain tissue      0.010437812  0.0022142489   4.713929  6.465203e-06
myCats.4.catsImmune            0.003461819  0.0014119087   2.451872  1.561547e-02
```

To assess the enrichment against the genomic background, we've extended **Figure S7** (called S5 in the initial submission) to also show the enrichment against the genomic background. For this figure, we added the following text to the **Online Methods**:

“... The enrichment against the genomic background was calculated based on the overlap per base pair as:

$(\text{intersectionOfOurAtacSeqPeaksetAndRoadmapData}/\text{sizeOfRoadmapData})/(\text{sizeOfOurAtacSeqPeakset}/\text{genomeSize}),$

where the *genomeSize* is the non-gapped genome (e.g. excluding centromeres and gaps in the assembly). *RoadmapData* is the actual DNase peaksets/chromHMM overlap shown in the respective figures. As the intersections concern thousands of peaks, randomness contributes little to the degree of enrichment. To substantiate this we redid the overlap calculations, but before this, each peakset (Neuron, non-Neuron, Glutamatergic, GABAergic, Oligodendrocytes, and Microglia/Astrocytes) was shuffled across the non-gapped genome with *BedTools*' "shuffleBed". Note that enrichments calculated against the genomic background are slightly biased due to non-uniform mappability and possibly other technical factors.”

Figure S7b is shown below. Note the results from the shuffling in the legend text.

Figure S7b

Figure S7. Overlap of Identified cell-specific open chromatin regions with existing epigenomic annotations.

The cell-specific OCRs were identified as detailed in the main text. The overlap was calculated by (a) the Jaccard index of the base pair overlap and (b) enrichment against the genomic background. Samples from REMC were aggregated into 4 groups: brain tissue, brain derived cells, immune cells/tissue, and other non-brain cells/tissues. The enrichment in panel b was also recalculated after shuffling the peaks. For the 2,286 overlap comparisons jointly assessed in this manner, the median enrichment was 0.97 (min 0.69, max 1.16, interquartile

range 0.95-1.00). Note that the REMC open chromatin/chromHMM states are all identified in the given cell/tissue and not those specific to it. Dots near the boxes represents outlier samples.

It is not unexpected that all of the plots below show an enrichment against the genomic background above one, as 1) the imputed Roadmap DNase/chromatin states are not differential DNase/chromatin states but all regions identified in a given cell type in their imputed dataset. 2) Comparing against the genomic background is slightly biased as noted before due to technical artefacts such as mappability. The latter bias can also in part be seen in the randomly shuffled analysis with the median being 0.97 and not 1.

We have added the following statement to the main text: “Comparing to the genomic background, our cell specific OCRs showed 5-27 fold enrichments in the brain related REMC DNase samples (**Figure S7b**).”

Jaccard Indices and enrichment metrics offer complementary information. Jaccard weighs the relative sizes of the annotations, which is a goal in **Figure 2b**. The enrichment could be marked in one comparison but with one of the annotations being quite small and thus be based on a comparatively small part of the genome. We would thus prefer to keep Figure 2b in the main paper, but we have created an alternative version called “Alternative Figure 2b” below. All of the information in “Alternative Figure 2b” is also found in **Figure S7b**.

Alternative Figure 2b

6. The enhancer construct analysis in specific cell types is a nice addition to the paper. However, the results are difficult to interpret. Why does only cortical layer 5 neurons express glutamate cell-specific enhancer (BDNF enhancer) and why is there a relatively low colocalization between the GABA cell-specific enhancer (DLX6), while the MGAS cell specific enhancer (TYROBP) is not observed in MGAS cells? Could there be species differences that affect the interpretability of the results? Would in vitro/human cell culture experiments have been a better choice? If enhancer constructs are a concern (as mentioned in the discussion) analysis of cell-type specific Hi-C data could be an option.

Response:

We also noted the varying functionality, or validation, of the putative enhancers, and in the original discussion, we included this paragraph:

Possible explanations for the lack of validation for the MGAS (TYROBP) OCR include: (1) interspecies differences; (2) specific promoter-enhancer interactions are lost when only transfecting the putative enhancer; (3) conformational/insulator differences between the in-situ enhancer and the transgenic mice; (4) uncertainty of the original OCR function; and (5) insertional effects overcame the CTCF insulators in the vector.

In response, we will elaborate on several of these points:

OCRs have proven to not all be active enhancers, and in fact, multiple validation papers show only 30-50% will successfully validate. In addition to what we included in the first version of the manuscript, other reasons include the need to act in a combinatorial manner and the fact that some OCRs are actually repressive.

Regarding interspecies differences, many enhancers, particularly developmental, are highly conserved between species (doi: 10.1016/j.cell.2013.05.056; doi: 10.1038/nature08451; doi: 10.1038/nature05295), and even if a particular sequence acts as an enhancer only in human, the appropriate binding factors are usually present in the mouse. We certainly considered in vitro/human cell culture systems for validation, but ultimately decided that none of the systems would ensure that the cells would be mature enough considering the OCRs were identified in adult brain, and also, there would be no spatial resolution.

We hoped that most interspecies differences would not preclude validation via mouse transgenesis, but this possibility cannot be fully discarded.

Hi-C data would of course be useful for interpretation of the promoter interactions of the OCRs we chose to validate, but since we speculate that cell-type specific interactions are occurring, no public data was available during the construction of our vectors. The putative enhancer selected for GABA population, was identified as a GABA-specific peak by H3K27ac ChIP-seq performed in DLPFC (doi: 10.1126/sciadv.aau6190). We assume that the variability of DLX6 enhancer is more susceptible to insertional effects compared to the others. Very recently, PLAC-seq analysis performed in the cortex of human individuals, revealed cell-type specific interactions between OCRs (doi: 10.1126/science.aay0793). The authors identified the same OCRs that we used to validate the TYROBP enhancer as a cell-type specific enhancer.

In this paper we have examples of each of the previously discussed points; (1) CNBP single enhancer showed activity within the expected cell population, (2) BDNF was specific for only one layer of the cortex, suggesting that other BDNF enhancers exist, (3) DLX6 was partially validated but with differences between founders and (4) TYROBP although meeting all criteria is likely not an active enhancer or belongs to some distant gene with this strange pattern of expression.

7. The use of LD score regression to examine the overlap of disease risk SNPs is nicely done. However, the diseases selected (neuropsychiatric diseases) are rather expected to be associated with neuronal OCRs based on several other publications (i.e. Price et al. 2019 Genome Biology, Polioudakis et al., 2019 Neuron). Could it be more interesting to examine overlap with the 170+ diseases in the LDscore regression tool with the

brain cell types in this study? This could generate new leads on the contributions of these cell type to other diseases.

Response:

We agree completely with the reviewer that broad application of LDSc-regression to assess the cell types implicated in various traits is of interest. And we acknowledge that the two referenced studies are also nice ones. There are no predefined list of traits used for the LDSc-regression, and all traits where the summary statistics are attainable could potentially be used as an input for the program (e.g. LD hub has aggregated >800 GWASs), though each file needs to be manually obtained and wrangled into the format expected by the program.

In our study, however, we would have problems with a too high burden of multiple testing. Knowing our annotation to be relatively scarce compared to the the broader regions and more powerful peaksets identified by ChIP-seq (PMID: 26414678, PMID: 25693563), and testing against the general genomic background of the 53 element baseline model, we focussed on well powered neuropsychiatric traits along with a few traits not expected to involve the brain as negative controls.

As a compromise between scope and burden of multiple testing, we've added 16 additional mostly brain related traits bringing the total to 30. This means we are on par with Price et al. and above Poliudakis et al. in terms of number of traits tested.

For more general testing of neuronal epigenomics broadly against GWAS traits we would await ChIP-seq epigenomic data or more powerful approaches than LDSc-regression (which for instance only considers variants with a MAF>5%).

Minor Issues

1. The three brain regions studied should be mentioned in the abstract

Response:

We agree and have now added the three brain regions to the abstract.

2. The abstract states that glutamatergic neurons are the most affect cell type in schizophrenia - this is a somewhat misleading statement because though true among the cells studied in this paper, there was no analysis of dopaminergic cell type which likely also have a strong role in this disease.

Response:

We agree that this limitation should be stated and have now rephrased the sentence in the abstract as: "Combining our cell specific open chromatin with a bulk tissue study of schizophrenia brains increased statistical power and confirmed glutamatergic neurons to be the most affected amongst the four assayed cell types.". We've further added the following to the discussion: "It would be interesting to assay the epigenome of dopaminergic neurons and particularly their involvement in schizophrenia. We do, however, also note that our study encompassed the DLPFC, which has long been implicated in schizophrenia (PMID: 3382321) and does not particularly relate to dopaminergic function."

3. The first sentence of the second paragraph in the introduction need to be changed to clearly explain the limitation in the field.

Response:

We apologize for the lack of clarity here. The sentence used to be “Studies of the human brain epigenome have, however, focused mostly on bulk tissue, in vitro cultured cells, included only two broadly defined brain cell types (neurons and non-neurons), or were performed in a single brain region²⁻⁴”, and have now been rephrased as: “The epigenome of the human brain is, however, still poorly understood, in part due to technical limitations. Even in fresh tissue, which is not readily available, isolation of intact cells is challenging. Although promising, the use of iPSC derived brain cells and/or organoids are not ideal proxies. Frozen archival tissue is more easily available, but the majority of cell type specific markers are lost upon thawing. Still, nuclei can be extracted and identified using a currently limited set of antibodies specific to the nucleus of a given cell type. Thus, studies have previously been limited to examining bulk tissue, in vitro cultured cells, included only two broadly defined brain cell types (neurons and non-neurons), or were performed in a single brain region²⁻⁴”.

4. For bioinformatic outlier removal, please provide a read or peak coverage and/or correlation value threshold.

Response:

We have changed the underlined text in the supplement: “We examined libraries that had a low FRiP (<5%), had a low final read count (<5 million reads), visually were outliers in clustering, or looked to have outright failed when inspecting the bigWig track. In this analysis, one sample with GABAergic neurons from the anterior cingulate cortex was clearly of inferior quality and was left out, thus leaving 47 samples for downstream analyses”. FRiP, or “fraction of reads in peaks”, is basically a signal to noise ratio, and based on the Bayesian Information Criterion analysis, the most important one in our analysis. As FRiP improved the model the most, it was added to the model in the first iteration. In the next iteration no other covariate was estimated to improve the fit of the model. In other words, FRiP was the most important technical covariate of those that we tested. We see this as a justification of using FRiP in the outlier identification step. A limitation to FRiP, as with many other quality metrics, is that it depends on other factors than strictly quality. In this case peak calling would directly affect FRiP, as more peaks/larger coverage would mean a higher fraction of reads would be within the peaks.

5. The authors should mention the age and postmortem interval of the samples. Perhaps this could contribute to the 34% residual value in Fig 1f.

Response:

We have now added the following to the main text “...of four individuals in early adulthood (ages 20-28), who had not been diagnosed with neuropsychiatric illness at the time of death, and all with a post-mortem interval less than 24 hours.” Unfortunately, we don’t have more specific times for the post-mortem intervals.

It is a valid point that such factors could contribute to the residuals. We have added the following section “Sampling error from finite sampling depth and untested technical confounds likely contributes to the residuals in the model. This includes covariates relating to the individual person such as post-mortem interval, but as

such covariates would show collinearity with "Person" in the model, which explains only a modest fraction of the variance, such person-related covariates are unlikely to drastically affect chromatin variability." In addition, from ongoing studies from our large group we have found PMI to have little effect or no effect at all compared to other covariates such as FRiP. This mirrors RNAseq analysis where PMI will have a small effect, especially compared to other variables (such as RIN and pH) that seems to capture better the quality of brain tissue. Although age as a covariate is expected to have a large effect on chromatin accessibility, this effect is expected to be minimum or have no effect at all given the limited age range of our cohort.

6. Promoter accessibility changes with both brain region and cell type. Was this taken into account during the overlapping analysis?

Response:

In short, yes, we do account for both cell type and brain region. We apologize that this wasn't written more clearly in the original submission. We have added the following to the methods section: "We here thus model both cell type and brain region effects." As well as: "In each of these contrasts, the cells were compared with the respective cells of the same brain region, and significance was established as this contrast differing from 0 (e.g. $p(\text{GLU_ACC} - \text{GABA_ACC} + \text{GLU_DLPFC} - \text{GABA_DLPFC} + \text{GLU_PVC} - \text{GABA_PVC} \neq 0)$). In this way potential overall differences in brain region are accounted for."

7. Why is the protein coding gene map comparison to GREAT in cell-specific pathways enrichments not provided as a supplementary figure (line 211)? Should a correlation analysis be included?

Response:

We apologize for not detailing this further. This approach seemed less powerful, and is not well established like GREAT. In particular, functionally related genes tend to be aggregated across the genome, and this approach might be slightly biased towards picking up gene sets, which have genes mostly in gene dense regions. Therefore, we would like to omit the details of these results from an already long paper to avoid confusion with the main results. We have added this sentence "This alternative approach to gene set enrichment analyses, however, seemed less powerful." to the paper. For the reviewers and the journal, the top pathways and p-values using this alternative approach is shown in the table below, but we would prefer not to include it:

Cell set	Pathway	P-value (BH-adjusted)
GLU	Signaling By GPCR (Re)	5.3E-27
GLU	Olfactory Transduction (KG)	2.9E-21
GLU	Sensory Perception (GO)	3.7E-21
GLU	Peptide Cross Linking (GO)	1.4E-13
GLU	Keratinocyte Differentiation (GO)	8.6E-12
GABA	Central Nervous System Development (GO)	6.7E-02
GABA	Cognition (GO)	6.7E-02
GABA	Reg. of Hormone Levels (GO)	6.7E-02
GABA	Neuron Differentiation (GO)	1.0E-01
GABA	Memory (GO)	1.1E-01
OLIG	Gliogenesis (GO)	7.6E-03
OLIG	Oligodendrocyte Differentiation (GO)	3.4E-02
OLIG	Peripheral Nervous System Development (GO)	4.2E-01
OLIG	Glial Cell Development (GO)	6.1E-01
OLIG	Ensheathment of Neurons (GO)	9.5E-01
MGAS	Neg. Reg. of Transcription From RNA Polymerase II Promoter (GO)	3.9E-09
MGAS	Response to Wounding (GO)	3.1E-07
MGAS	Neg. Reg. of Cell Proliferation (GO)	3.8E-07
MGAS	Epithelium Development (GO)	5.1E-07
MGAS	Neg. Reg. of Multicellular Organismal Process (GO)	6.3E-07
Neuron	Olfactory Transduction (KG)	2.6E-15
Neuron	Sensory Perception (GO)	3.5E-14
Neuron	Signaling By GPCR (Re)	1.6E-09
Neuron	Neuroactive Ligand Receptor Interaction (KG)	1.6E-06
Neuron	Cell Cell Signaling (GO)	4.3E-06

non-Neuron	Neg. Reg. of Transcription From RNA Polymerase II Promoter (GO)	1.9E-06
non-Neuron	Pi3kci AKT Pathway (PI)	3.8E-03
non-Neuron	AKT Phosphorylates Targets in The Cytosol (Re)	7.1E-03
non-Neuron	Transcription From RNA Polymerase II Promoter (GO)	7.1E-03
non-Neuron	Covalent Chromatin Modification (GO)	7.1E-03

8. Figure 3b does not clearly show the overlap of TFs from multiple cell types. For example, the oligodendrocyte lines are over the non-neuronal lines making it difficult to see the overlap in TF like SF family. Separate cell type plot or modifying the existing plot could improve the clarity of this figure.

Response:

We acknowledge this is a problem, and have addressed the problematic figure by nudging the connections so they are more clearly visible when more than one annotation connects to the same transcription factor.

9. Multiple figures have axis too small to effectively read, making it difficult to interpret data. Please edit all figures accordingly.

Response:

We acknowledge this and apologize. Figures 1, 2, 5, S6, S8, S7, and S9 have been tweaked.

Reviewer #2 (Remarks to the Author):

The study by Dr. Roussos and colleagues carries out ATAC-seq on 3 cerebral cortex regions in 4 individuals using FANS for NeuN, Sox6 and Sox10 to delineate 4 distinct cell types GABA neurons, Glu neurons, oligodendrocytes and microglia/astrocytes. This study follows up on the 2018 Genome Research paper of the same group which characterised NeuN+ and NeuN- cell populations, by further refining cell types.

The study is thorough, well written, and carries out standard analyses of ATAC-seq data, such as overlap analyses with epigenomic features, TSS, TF binding sites etc., as well as assessment of enhancer function in vivo for several open-chromatin regions.

Major Issues

1. The QC FRiP score are quite low: 0.05-0.15, while the standard for Encode data for example is >0.2. The authors need to provide additional information to demonstrate an acceptable signal-to noise ratio, such as, for example a TSS enrichment score (<https://www.encodeproject.org/data-standards/terms/#enrichment>) and a replication rate or a correlation across replicates (i.e. different individuals , same brain region and cell type).

Response:

We appreciate the reviewer's concern about data quality. We note that our experiments were performed on nuclei extracted from frozen postmortem brain specimens. Few studies have applied ATAC-seq to frozen tissue. Such tissues are, by their very nature, suboptimal due to both the freeze/thawing and the post mortem interval. This would be expected to have a direct and adverse effect on data quality when compared to fresh tissue/cell lines. As evidenced by the results presented in the paper or a simple inspection of the UCSC tracks, however, the ATAC-seq data very much still provide valuable insights into the cell specific human brain epigenomes.

To more quantitatively assess the concerns about the quality of our data, we processed three publicly available ATAC-seq datasets through our pipeline:

- Fullard et al 2018 (115 NeuN+/- samples from 14 regions; PMID: 29945882)
- Bryois et al (248 bulk tissue samples from DLPFC; PMID: 30087329)
- Rizzardi et al (22 NeuN+/- samples from DLPFC and NUC; PMID: 30643296)

Using the *ataqv* package (PMID: 32213349) to calculate TSS enrichment, and found our data to have comparable levels of TSS enrichment as shown below (**NEW FIGURE 1b**). This is despite the fact that we employ the most challenging experimental protocol as we are assaying four brain cell subtypes, not only NeuN+/- or bulk tissue. TSS enrichment metric for all samples of this study was added to **Table S1**.

To study the reproducibility among the biological samples within the same cell type and brain region, we performed pairwise correlation by estimating the overall similarity between the raw reads coverage (number of reads) over consecutive bins of 10,000 bp genomic regions (**Figure NEW FIGURE 1a**).

Both figures have now been included in the supplement along with the explanation in the **Online Methods**.

NEW FIGURE 1. Intersample correlation and TSS enrichment. (a) Correlations of raw reads counts over consecutive bins of 10,000 bp genomic regions between samples originating from the same cell type and brain region. (b) Comparison of TSS enrichment in housekeeping genes for our dataset and three additional

postmortem human brain ATAC-seq datasets. Note that the double sorting employed in the current study puts more stress on the nuclei than a single sorting step (Fullard and Rizzardi) or simply using bulk tissue (Bryois).

2. A comparison between the current ATAC-seq data and RNA-seq data from NeuN+/NeuN- cells (eg. PMID 31288836) is necessary for a functional interpretation of the open chromatin regions, rather than just overlaps (with gene sets from the Epigenomics Roadmap consortium).

Response:

We acknowledge the reviewer's request for comparisons against additional epigenomic and transcriptomic studies that assayed postmortem human brain tissue. To quantify the concordance of our differential analysis results (changes on ATAC-seq & RNA-seq level between neurons and non-Neurons), we compared our dataset to the suggested study as well as three additional ones:

- *RNA-seq from Mendizabal et al. 2019 (89 NeuN+/Olig2+ DLPFC samples; PMID:31288836)*
- *RNA-seq from Rizzardi et al. 2019 (20 NeuN+/- samples from 2 regions; PMID: 30643296)*
- *ATAC-seq from Rizzardi et al. 2019 (22 NeuN+/- samples from 2 regions; PMID: 30643296)*
- *ATAC-seq from Fullard et al. 2018 (115 NeuN+/- samples from 14 regions; PMID: 29945882)*

*Here, we calculated the correlation of our chromatin accessibility changes between neurons and non-Neurons with their differential analyses (**NEW FIGURE 2**). We observed very strong correlation with two epigenomics datasets (Pearson correlation of 0.934 and 0.894), implying a robustness of our findings. Transcriptomics datasets showed lower but still very convincing levels of concordance (Pearson correlation of 0.491 and 0.466), especially considering our simplistic approach utilizing only promoter OCRs (**Online Methods**). These results have been added to the manuscript.*

NEW FIGURE 2. Comparison of cell specificity across RNA/ATAC-seq studies. The cell specificity of OCRs and gene expression was assessed using fold change concordance (log2 values) between our ATAC-seq dataset (fold changes for comparison between neuronal vs non-Neuronal / Oligodendrocyte samples) and external ATAC-seq / RNA-seq datasets. For external ATAC-seq datasets, correlation was calculated on overlapping OCRs (minimum 25% overlap). For external RNA-seq datasets, correlation was calculated between fold change of chromatin accessibility of promoter OCRs from our ATAC-seq and fold change of gene expression of corresponding genes from external RNA-seq study. This simplistic approach to correlating ATAC-seq to RNA-seq is not expected to yield high correlations, as distal elements are ignored and as OCRs include both enhancers and silencers. (a) ATAC-seq of NeuN+/- samples from 14 brain regions. (b) ATAC-seq from NeuN+/- samples from dorsolateral prefrontal cortex and nucleus accumbens. (c) RNA-seq from NeuN+/Olig2+ samples from dorsolateral prefrontal cortex. (d) RNA-seq from NeuN+/- samples from dorsolateral prefrontal cortex and nucleus accumbens.

3. The Jaccard index is not appropriate for calculating the overlap with different epigenomic features, as it does not take into account genome-wide background. A test with random sampling of the genomic background for each chromHMM category should be used instead.

Response: This has been addressed in the response to reviewer #1's 5th major issue.

4. For the variant enrichment analyses, an enrichment of eQTL data (eg. GTEx) would be valuable to demonstrate the enrichment of regulatory variants in the OCRs identified in the present study.

Response: We acknowledge the relevance and have now done an overlap analysis, which found SNPs likely to causally affect gene expression based on GTEx data to be enriched in the open chromatin regions. In particular, looking at the ratio of causal SNPs in the open chromatin regions compared to the background, odds ratios from 1.6 to 2.9 were observed. These results are shown in the new **Table S2** in the manuscript. We have added one author to the paper (Dr. Biao Zeng), who was responsible for this analysis.

5. The title is an overstatement: there is no assessment of the function of glutamatergic neurons and no experimental evidence for genetic variants associated with schizophrenia affecting such function.

Response:

We acknowledge this and have changed the title to: "Common schizophrenia risk variants are enriched in open chromatin regions of glutamatergic neurons".

Minor:

FigS6,7: It's unclear what Log2Fc refers to, as these are enrichment analyses.

Response:

We apologize for the lack of clarity, and have renamed it as log2 enrichment. We previously just had a citation for the methodology in the online methods, but have now added this text for clarification "In short, the number of OCRs overlapping the presumed regulatory domains of genes in a particular gene set is compared to OCRs overlapping any regulatory domains. Enrichment is then the OCR density for the regulatory domains of the gene set compared to the OCR density in the union of all regulatory domains."

Reviewer #3 (Remarks to the Author):

This paper provides the first ATAC-Seq study to examine subtypes of neurons and non-neuronal cells. Therefore, although the sample size is small I believe it is a strong publication (indeed, it provides a further example of the power of within-subjects analyses of this type). The authors should be commended for making their data open access, and for providing it for review - this greatly enhances both the potential impact of the paper and the ease of review. The manuscript is generally well written and the figures and tables are largely clear. I therefore have only questions of clarification:

1. On p7, and in Fig 2b, I believe that the REMC comparisons would benefit from clarification. For example, the text states “[...] our 160 cell-specific OCRs predominantly overlap with REMC brain samples” Is this a typo? If not, please clarify how OCRs overlap with samples. Similarly, in Fig 2b, please state what the asterisks represent (and clarify why some bars have multiple asterisks).

Response:

We apologize for the lack of clarity in the sentence in question and have rephrased it as follows: “To investigate how our data compared to existing epigenomic data, we computed the overlap of our OCRs with open chromatin from DNase-seq as well as chromatin states from the Roadmap epigenomics mapping consortium^{2,9} (REMC) (Figures 2b and S5). In terms of the Jaccard index, open chromatin and active chromatin states identified in REMC brain related samples showed a higher overlap with our cell specific OCRs than non-brain related samples.”

We apologize for not stating what the dots near the boxes in Figure 2b means. They are simply outlier samples identified by the plotting software (ggplot2 in R). This is now stated in the paper.

2. Table 1 also requires clarification: Table 1: where were the neuronal and non-neuronal data in this table obtained from? If they are simply the Glu + GABA and OLIG + MGAS why do the values not add up?

Response:

Table 1 is based on the identification of cell specific OCRs. In our admittedly quite lengthy Online Methods under “Statistical analysis of differences in chromatin accessibility” there is a very technical description of why neurons are not simply GLU+GABA etc. Here, we have expanded this for clarity: “...Resultantly, such cell specific OCRs are truly specific to the cell type in question. For example, an OCR showing high and comparable chromatin accessibility in GLU and GABA and a low accessibility in OLIG and MGAS would not be assigned as specific to GLU.”

A bit later we have added: “These “Neuron” and “non-Neuron” sets thus also include OCRs that are specific to the overall cell group but might be equally accessible in both of the constituent cell subtypes. For instance, an OCR which is highly and equally accessible in GLU and GABA but lowly accessible in both OLIG and MGAS would be listed here as a “Neuronal” OCR. It would, however, not be listed as GLU or GABA specific as detailed above.”

We have further added this to the legend of Table 1: ““Neuronal” is not simply the sum of “Glutamatergic” and “GABAergic”, as the two latter exclude chromatin that are not specific to either neuronal subtype (Online Methods). Likewise, “non-Neuronal” is not the sum of the two constituent cell subtypes.”

3. Regarding the GWAS-related findings, were results corrected for between-phenotype correlations? Furthermore, it would be good to add comment about the diagnostic categories that *don't* show clear association, e.g. bipolar. Furthermore, what corrections (if any) were run for multiple comparisons for these studies?

Response:

The way that LDSc is implemented, one GWAS at a time is tested against the annotations. Thus, each GWAS was run individually through the LDSc software. We apologize for this not being clear and have now elaborated on it in the Online Methods.

We have addressed some of the limitations concerning the power of LDSc in our response to Reviewer #1's, 7th major point. We have further added this comment: "Some neuropsychiatric traits did not show enrichment in our open chromatin regions, which might result from a lack of power in the GWAS, lack of power in the LDSc approach, or the limited genomic extent of our epigenomic annotations."

We used the "Benjamini & Hochberg" FDR correction for multiple testing, which has now been explicitly stated at Figure 5 and S17. We apologize for this not being the case in the initial submission.

4. It would be good to have more information about the postmortem brains included here pH, and ideally RIN. I realise that this study did not examine RNA directly, but these measures still speak to overall tissue quality. It might be that some of this information is in Table S1, but the table is only partially readable as PDF (e.g. what was the cause of death of Female sample #4)

Response:

*Unfortunately, neither RIN nor pH was available for our samples. However, we can get an idea about the impact differences in the sample quality across our four brains by looking at our variance partitioning analysis (**Figure 1f**). Cell and brain region contribute massively (57%) to the variance in chromatin accessibility, whereas "Person", representing the four different brain samples, had a comparatively low impact (1.5%). We believe "Fraction of reads in peaks (FRiP)" in table S1 to be a good but imperfect indicator of sample quality.*

We apologize for the cause of death for Female sample #4 being cut off in the files generated by the online manuscript system. It should of course be plainly visible in the final Excel file. The cause of death was "sudden natural, heart".

Minor point:

Why did the authors use Hg19 for their annotations? Could they provide a liftover to Hg38?

Response:

Hg19 was the prevailing assembly at the time of initial data preparation, and the existing external data used in the paper is based on hg19. We acknowledge that hg38 is now widely adopted, and we have therefore added hg38 peaks to GEO record (<https://www.ncbi.nlm.nih.gov/geo/query/acc.cgi?acc=GSE143666> - reviewer's password yfuzmmwwjbjvct). We have also updated UCSC open chromatin tracks available from the project webpage to support both hg19 and hg38 (<https://icahn.mssm.edu/boca2>).

REVIEWERS' COMMENTS:

Reviewer #1 (Remarks to the Author):

The authors have done a nice job answering the reviews and improving their manuscript. In particular they now present their findings in a manner that better highlights the novelty and applicability of their ATAC-seq resource and clarified methodological concerns. Their revised Figure 2b is appropriate. I have no further recommendations.

Reviewer #2 (Remarks to the Author):

The authors have satisfactorily addressed all of my questions. Thank you.

Reviewer #3 (Remarks to the Author):

I thank the authors for their thorough responses to my comments (and those of the other reviewers). They have satisfactorily addressed the points that I made.